# ALIGNMENT WITH RANKING AND RATING INFORMATION

## ABSTRACT

The class of direct preference optimization (DPO) algorithms has emerged as a promising approach for solving the alignment problem in foundation models. These algorithms work with very limited feedback in the form of pairwise preferences and fine-tune models to align with these preferences without explicitly learning a reward model. While the form of feedback used by these algorithms makes the data collection process easy, its ambiguity in terms of the quality of responses has significant negative implications, including incentivizing policies that favor out-of-distribution responses, a phenomenon referred to as *likelihood displacement*. In this paper, we study how DPO-style algorithms can leverage additional information in the form of *rating gap* which informs the learner how much the preferred response is better than the rejected one. We present new algorithms that can achieve faster statistical rates than DPO in presence of accurate rating gap information. Moreover, we theoretically prove and empirically show that the performance of our algorithms is robust to inaccuracy in rating gaps. Finally, we demonstrate the solid performance of our algorithms in comparison to a number of DPO-style algorithms across a wide range of LLMs and evaluation benchmarks.

## 1 INTRODUCTION

Learning from preference data (ranking information) has become a popular paradigm for solving the alignment problem in large language models (LLMs) (Christiano et al., 2017; Rafailov et al., 2023; Zhu et al., 2023). Although data collection is easier and the annotators' feedback is less noisy in this setting, compared to the rating style feedback, the amount of information that can be extracted from such data is very limited. Indeed, given a prompt, a preferred and a dispreferred response, several scenarios are likely: both responses are of high/low quality or one is good and the other one is poor. However, the contrastive learning approach used by Direct Preference Optimization (DPO) style algorithms, such as DPO (Rafailov et al., 2023) and Identity-mapping Preference Optimization (IPO) (Azar et al., 2024), is only justified for the latter case. In the cases that the two responses have high/low quality, it makes sense to imitate/forget both of them. However, it is not possible for a fine-tuning algorithm to identify which of these cases it is facing from the preference style information. The lack of information about the individual or relative quality of responses can create ambiguity for these algorithms and have negative implications on their performance. For example, it has been shown that it can incentivize in-sample probability reduction in DPO-style algorithms and create bias towards policies that favor out-of-distribution responses (e.g., Adler et al. 2024b; Xu et al. 2024b; Pal et al. 2024; Fisch et al. 2024; Xiao et al. 2024; Shen et al. 2024; Wu et al. 2024; D'Oosterlinck et al. 2024), a phenomenon referred to as *likelihood displacement* (Razin et al., 2025).

In this paper, we study a setting in which, in addition to preference/ranking feedback, the training dataset contains the relative rating of the responses, which we refer to as *rating gap*. We propose three algorithms derived from two different approaches that leverage this additional information in an efficient and principled way. We derive our first two algorithms, *Rating DPO* (RDPO) and *Rating IPO* (RIPO), by changing the RLHF objective to maximize a linear combination of the ranking and rating information. Our third algorithm, *Maximum-Likelihood-based Rating DPO* (ML-RDPO), is derived using the maximum-likelihood (ML) principle by making certain assumptions on the joint distribution of the ranking and rating information. In ML-RDPO, we consider a linear combination of the likelihood objectives rather than a linear combination of the two sources of information, ranking and rating, as is done in the derivation of RDPO and RIPO. We provide theoretical analysis for our

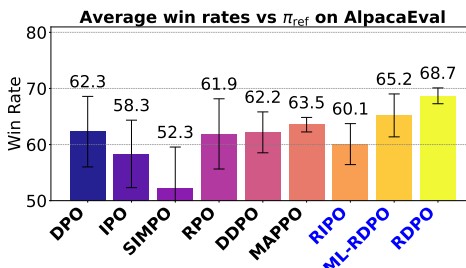 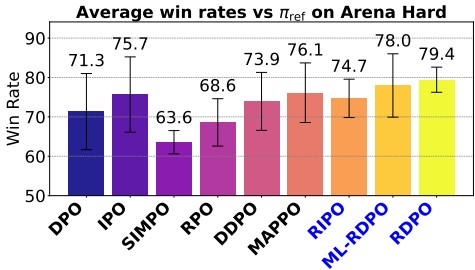

Figure 1: Win rates averaged for $\pi_{\text{ref}} \in \{\texttt{Llama3.1-8B}, \texttt{Zephyr-7B}, \texttt{Mistral-7B}\}$. Our methods are labelled in blue.

algorithms showing that they can achieve faster statistical rates than DPO in the presence of accurate rating gap information, and are robust in case this information is noisy. We empirically evaluate our algorithms and compare them to a number of DPO-style methods across a wide range of LLMs and evaluation benchmarks. The results validate our theoretical findings and demonstrate the solid performance of our algorithms.

**Theoretical Contributions.** We prove two desirable properties, *acceleration* and *robustness*, for our algorithms. Indeed, if the ratings are consistent with the latent reward model used by the annotator to rank the responses (under the Bradley-Terry formulation), RDPO and ML-RDPO can accelerate learning compared to DPO. That is, they are able to learn a well-performing policy faster (or equivalently a better policy given a dataset of the same size) than an algorithm that only leverages ranking information (e.g., DPO or IPO).

**Empirical Contributions.** In addition to these nice theoretical properties, RDPO and ML-RDPO perform well in practical alignment problems. Figure 1 provides a summary of our experiments (more experiments in Section 6 and the appendix), showing the average win-rate of the checkpoints fine-tuned by different algorithms against the reference model $\pi_{\text{ref}}$ in AlpacaEval (Li et al., 2023b) and ArenaHard (Li et al., 2024) benchmarks. The means and standard deviations are computed over 3 independent experiments where $\pi_{\text{ref}}$ is one of the following models: Zephyr-7B[1], Llama-3.1-8B[2], and Mistral-7B[3]. It can be seen that RDPO and ML-RDPO are the best and second best performing methods, respectively. Moreover, since RDPO has the smallest variance, it can be considered as the most consistently well-performing method across the choices of $\pi_{\text{ref}}$ we experimented with.

In agreement with the theoretical findings, RDPO and ML-RDPO outperform both ranking-only methods, such as DPO (Rafailov et al., 2023), SIMPO (Meng et al., 2024), and IPO Azar et al. (2024), and rating-only methods, such as Distilled DPO (Fisch et al., 2024). Moreover, their performance is superior to RPO (Adler et al., 2024a) and MAPPO (Lan et al., 2025), which are recent alignment methods leveraging both rating and ranking information, as done by RDPO and ML-RDPO. Finally, while RIPO does not perform as well as RDPO in the experiments of Figure 1, it still performs well in some of the experiments we report in Section 6 and the appendix. Moreover, it serves as an example that our principled derivation of RDPO can be extended beyond the cases where the Bradley-Terry assumption holds.

**Paper Organization.** In the following section, we formalize the alignment problem. Then, in Section 3, we derive RDPO, RIPO, and ML-RDPO from first principles. Finally, we present their theoretical guarantees in Section 4, discuss the closest related work in Section 5, and demonstrate their practical effectiveness in Section 6 before concluding the paper in Section 7.

## 2 PRELIMINARIES

We present the main ingredients of the alignment problem in LLMs on which we will build in the subsequent sections. The input to most alignment algorithms is a dataset $\mathcal{D}$ of tuples $(x, a, a', z)$,

---

[1] https://huggingface.co/alignment-handbook/zephyr-7b-sft-full
[2] https://huggingface.co/allenai/Llama-3.1-Tulu-3-8B-SFT
[3] https://huggingface.co/HuggingFaceH4/mistral-7b-sft-beta

where $x$ is a prompt sampled from a distribution $\nu_0$, $a$ and $a'$ are two responses, and $z$ is a Bernoulli random variable that is one if the response $a$ is preferred to $a'$ conditioned on $x$, i.e., $a \succ a' \mid x$ and is zero, otherwise. Given $z$, we define $a^+ = a$ and $a^- = a'$ if $z = 1$ and viceversa $a^+ = a'$ and $a^- = a$ if $z = 0$. The binary variable $z$ is assumed to be sampled according to the Bradley-Terry (BT) model[4] (Bradley & Terry, 1952), i.e., $z \sim \texttt{Bern}\big(\sigma(r^\star(x,a) - r^\star(x,a'))\big)$, where $\sigma(x) = 1/(1+e^{-x})$ is the sigmoid function and $r^\star$ is the latent reward model (RM) of the annotator. We are also given a reference policy $\pi_{\text{ref}}$ (often the SFT checkpoint $\pi_{\text{SFT}}$) which serves as a guardrail.

In RLHF, we first employ $\mathcal{D}$ to train a parameterized RM, $r_\phi$, and then use it to solve the following KL-regularized RL problem:

$$\max_\theta \ \mathbb{E}_{x \sim \nu_0}\Big[\mathbb{E}_{a \sim \pi_\theta(\cdot|x)}\big[r_\phi(x,a)\big] - \beta \cdot \mathbb{KL}\big(\pi_\theta(\cdot|x)\|\pi_{\text{ref}}(\cdot|x)\big)\Big], \tag{1}$$

where $\beta > 0$ is a hyper-parameter denoting the relative importance of reward maximization against ensuring a low deviation from $\pi_{\text{ref}}$. The RM is learned by minimizing the cross-entropy (CE) loss

$$\min_\phi \sum_{(x,a^+,a^-) \in \mathcal{D}} -\log \sigma\big(r_\phi(x,a^+) - r_\phi(x,a^-)\big). \tag{2}$$

Fine-tuning $\pi_\theta$ in the RLHF approach is split into two stages: reward learning using the BT model, followed by a policy optimization using Eq. 1. More recently, a family of algorithms has emerged that solves the above two problems in a single stage, without learning a RM explicitly. The most notable among this family is DPO (Rafailov et al., 2023). The key insight here is that given a RM $r$, the problem in Eq. 1 admits the following closed-form solution:

$$\pi_\beta^\star(a|x) = \pi_{\text{ref}}(a|x)\exp(r(x,a)/\beta)/Z(x), \tag{3}$$

where $Z(x)$ is the partition function. The next step is to introduce a parameterization of the reward $r$ by first parameterizing the closed form solution $\pi_\beta^\star$ as $\pi_\theta$ in Eq. 3, and then inverting it for $r$ as

$$r_\theta(x,a) = \beta \cdot \log \frac{Z(x)\pi_\theta(a|x)}{\pi_{\text{ref}}(a|x)}. \tag{4}$$

The final step is to substitute the parameterized reward from Eq. 4 into the CE-loss in Eq. 2, the partition function $Z(x)$ is canceled out, leading to the following loss for DPO:

$$\mathcal{L}_{\text{DPO}}(\theta) = \sum_{(x,a^+,a^-) \in \mathcal{D}} -\log \sigma\big(\beta \cdot \Delta_\theta(x,a^+,a^-)\big), \tag{5}$$

where $\Delta_\theta(x,a^+,a^-) = \log \frac{\pi_\theta(a^+|x)}{\pi_{\text{ref}}(a^+|x)} - \log \frac{\pi_\theta(a^-|x)}{\pi_{\text{ref}}(a^-|x)}$.

Another popular member of the family of DPO-style algorithms is IPO (Azar et al., 2024). IPO uses the following square loss to fine-tune the LLM $\pi_\theta$:

$$\mathcal{L}_{\text{IPO}}(\theta) = \sum_{(x,a^+,a^-) \in \mathcal{D}} \big(\Delta_\theta(x,a^+,a^-) - \frac{1}{2\beta}\big)^2. \tag{6}$$

We conclude the section, introducing some notation that will turn out useful in the rest of the paper.

**Notation.** We denote by $\mathcal{X}$ and $\mathcal{A}$ the set of prompts given to and the set of responses generated by a LLM. Moreover, we denote by $\Pi$ the class of policies that can be induced by a choice of the parameters $\theta$, i.e., $\Pi = \{\pi \mid \exists\, \theta \text{ s.t. } \pi(a|x) = \pi_\theta(a|x) \,\forall x, a \in \mathcal{X} \times \mathcal{A}\}$. We use $|\Pi|$ to show the size of this class.[5] Finally, we denote the expected reward or performance of a policy $\pi$ under reward $r$ as the following quantity $\langle \pi, r \rangle := \sum_{x \in \mathcal{X}} \nu_0(x) \sum_{a \in \mathcal{A}} \pi(a|x) r(x,a)$. Finally, we denote the optimal policy under the latent reward model $r^\star$ as $\pi^\star = \operatorname{argmax}_{\pi \in \Pi} \langle \pi, r^\star \rangle$.

---

[4]This can be extended to the more general Plackett-Luce ranking model (Plackett, 1975; Luce, 2012) if we have access to several ranked responses.

[5]For infinite classes, the result can be generalized using covering numbers.

# 3 ALGORITHMS FOR ALIGNMENT WITH RANKING AND RATING

We now consider the problem of fine-tuning a LLM where the original dataset is augmented with the *rating gaps* of the responses, i.e.,

$$\mathcal{D} = \left\{ (x_i, a_i, a_i', z_i, \hat{r}(x_i, a_i^+) - \hat{r}(x_i, a_i^-)) \right\}_{i=1}^N, \tag{7}$$

where $\hat{r}$ is an approximation of the latent RM of the annotator $r^\star$. Thus, the rating gap $\Delta_{\hat{r}}(x_i, a_i^+, a_i^-) := \hat{r}(x_i, a_i^+) - \hat{r}(x_i, a_i^-)$ serves as an estimate of how much better is the chosen action $a_i^+$ compared the discarded one, i.e., $a_i^-$. Having introduced the dataset, we are now ready to present our algorithms in the next sections.

## 3.1 RATINGS DIRECT PREFERENCE OPTIMIZATION (RDPO)

For deriving RDPO, we consider the RLHF objective in Eq. 1 where the reward of interest is a linear combination of the original reward model (RM) $r$ and the rating estimate $\hat{r}$, i.e.,

$$\max_\theta \ \mathbb{E}_{x \sim \nu_0} \left[ \mathbb{E}_{a \sim \pi_\theta(\cdot|x)} \left[ r(x,a) + \frac{\beta \hat{r}(x,a)}{\beta_1} \right] - \beta \cdot \mathbb{KL}\big(\pi_\theta(\cdot|x) \| \pi_{\text{ref}}(\cdot|x)\big) \right]. \tag{8}$$

The parameter $\beta_1$ weighs the contribution of the rating information $\hat{r}$ in Eq. 8. It is inversely related to our confidence in the accuracy of the rating information: the smaller $\beta_1$ is, the more confident we are about the accuracy of $\hat{r}$. The problem in Eq. 8 admits the following closed-form solution:

$$\pi_\beta^\star(a|x) = \pi_{\text{ref}}(a|x) \exp \left( \frac{r(x,a)}{\beta} + \frac{\hat{r}(x,a)}{\beta_1} \right) / Z(x). \tag{9}$$

Inverting the above equation and approximating $\pi_\beta^\star$ via $\pi_\theta$, we can define the following implicit reward model parametrized for any policy parameter $\theta$ as $r_\theta(x,a) = \beta \cdot \log \frac{Z(x)\pi_\theta(a|x)}{\pi_{\text{ref}}(a|x)} - \frac{\beta}{\beta_1} \cdot \hat{r}(x,a)$. Finally, plugging $r_\theta$ into Eq. 2, we obtain the following loss for RDPO:

$$\mathcal{L}_{\text{RDPO}}(\theta) = \sum_{i=1}^N -\log \sigma \left( \beta \Delta_\theta^i - \frac{\beta}{\beta_1} \Delta_{\hat{r}}^i \right), \tag{RDPO}$$

where for compactness we used $\Delta_\theta^i := \Delta_\theta(x, a_i^+, a_i^-)$ and $\Delta_{\hat{r}}^i := \Delta_{\hat{r}}(x, a_i^+, a_i^-)$. Following similar steps, we can derive several variants of RDPO as explained in the rest of this section.

**RDPO variants.** Replacing the BT preference model with more general alternatives, as in IPO Azar et al. (2024), we can derive Ratings IPO (RIPO) loss in which the log sigmoid function in RDPO is substituted with a parabola centered at $1/2$. That is,

$$\mathcal{L}_{\text{RIPO}}(\theta) := \sum_{i=1}^N \left( \beta \Delta_\theta^i - \frac{\beta}{\beta_1} \Delta_{\hat{r}}^i - 1/2 \right)^2. \tag{RIPO}$$

It is easy to see that the RIPO loss is a shifted version of the loss used by Distilled-DPO (Fisch et al., 2024), which, of course, was derived from a different perspective. Other RDPO variant can be derived by replacing the KL divergence in Eq. 8 with other $f$-divergences such as $\chi^2$.

We conclude this section by noticing that RDPO does not make any statistical assumption about the rating gap information. However, it requires access to the rating gaps $\Delta_{\hat{r}}(x, a^+, a^-)$ for all $(x, a^+, a^-) \in \mathcal{D}$ (not necessarily their actual ratings). In the next section, we propose ML-RDPO to bypass this requirement at the cost of assuming that $\Delta_{\hat{r}}(x, a^+, a^-)$ is a Gaussian random variable.

## 3.2 MAXIMUM-LIKELIHOOD-BASED RDPO (ML-RDPO)

In order to derive ML-RDPO, we consider a *linear combination of the log likelihood objectives* rather than a linear combination of the two reward models. To this end, for any reward model $r$, let $P_r^{\text{rat,rank}}$ be the joint probability distribution of the ranking and rating information. The idea behind the ML-RDPO algorithm is to output the policy $\pi_{\text{out}} \propto \pi_{\text{ref}}(a|x) \exp(\tilde{r}(x,a)/\beta)$ in which $\tilde{r}$ is an approximation of $r^\star$ computed maximizing the joint log-likelihood, i.e.,

$$\tilde{r} = \operatorname*{argmax}_{r \in \mathcal{R}} \ \log P_r^{\text{rat,rank}}(\mathcal{D}), \tag{10}$$

where $\mathcal{R}$ is parameterized by the policy weights as $\mathcal{R} = \{r \mid \exists\, \theta \text{ s.t. } \beta\Delta_\theta = \Delta_r\}$ where the equality holds elementwise. In order to compute $\pi_{\text{out}}$ efficiently without the need of computing $\tilde{r}$, we assume conditional independence and Gaussian distributed rating gaps, to simplify the joint log-likelihood as shown next in Lemma 3.1.

**Lemma 3.1.** *Let us assume that the rating gaps $\Delta_{\hat{r}}^i | x_i, a_i, a_i'$ are Gaussian distributed with variance $\mathbb{V}$ and conditional independence between $z_i$ and $\Delta_{\hat{r}}^i$ given $x_i, a_i, a_i'$ is satisfied. Then, it holds that*

$$\log P_r^{\text{rat,rank}}(\mathcal{D}) = \sum_{i=1}^{N} \log \sigma(\Delta_r^i) - (2\mathbb{V})^{-1}\left(\Delta_{\hat{r}}^i - \Delta_r^i\right)^2,$$

*where $\Delta_r^i := r(x_i, a_i) - r(x_i, a_i')$.*

Now, given the definition of the induced reward class $\mathcal{R}$, we can perform the change of variable from $r$ to $\beta\log(\pi_\theta/\pi_{\text{ref}})$ in the optimization problem in Eq. 10 to obtain

$$\min_\theta \; \mathcal{L}_{\text{ML}-\text{RDPO}}(\theta) := \underbrace{\sum_{i=1}^{N} -\log\sigma(\Delta_\theta^i)}_{\text{Ranking term}} + (2\mathbb{V})^{-1}\underbrace{\sum_{i=1}^{N}\left(\Delta_{\hat{r}}^i - \Delta_\theta^i\right)^2}_{\text{Rating term}}. \qquad \text{(ML-RDPO)}$$

The policy $\pi_{\text{out}}$ is therefore approximated via $\pi_{\theta^\star}$ where $\theta^\star$ is a minimizer of $\mathcal{L}_{\text{ML}-\text{RDPO}}$. Notice that $\mathcal{L}_{\text{ML}-\text{RDPO}}$ is the sum of a term involving only the ranking information and of a second one computed solely from rating. This allows to easily apply ML-RDPO even when the ranking and the rating information are not seen for the same prompts-responses tuples. Moreover, the ranking term coincides with the DPO loss and the rating term is instead the Distilled-DPO loss weighted by $\mathbb{V}^{-1}$. Therefore, the more confidence we have in the rating information, the smaller the value for $\mathbb{V}$ should be. At an intuitive level, $\mathbb{V}$ in ML-RDPO acts similarly to $\beta_1$ in RDPO.

## 4 THEORETICAL RESULTS

Our main theoretical contribution is to prove that augmenting the DPO dataset with ratings gap $\hat{r}(x_i, a_i^+) - \hat{r}(x_i, a_i^-)$ information allows to achieve better statistical guarantees. However, the benefit depends heavily on the quality of the rating estimates $\hat{r}$. In order to formalize this concept, let us introduce the approximation error

$$\text{Err}_{\pi_{\text{ref}}}(\hat{r}) = \mathbb{E}_{x\sim\nu_0}\mathbb{E}_{a,a'\sim\pi_{\text{ref}}(\cdot|X)}\left[\left(r^\star(x,a) - r^\star(x,a') - \hat{r}(x,a) + \hat{r}(x,a')\right)^2\right].$$

Intuitively, $\text{Err}_{\pi_{\text{ref}}}(\hat{r})$ quantifies how close $\hat{r}$ is to $r^\star$ under the support of the reference policy $\pi_{\text{ref}}$. Note that requiring small $\text{Err}_{\pi_{\text{ref}}}(\hat{r})$ is much weaker than requiring $\hat{r}(x,a)$ to be close to $r^\star(x,a)$ for all $x, a \in \mathcal{X} \times \mathcal{A}$. In practice, small $\text{Err}_{\pi_{\text{ref}}}(\hat{r})$ is a reasonable assumption for datasets like `ultrafeedback_binarized`[6] in which $\hat{r}$ is obtained querying a (rather) reliable grader such as Chat GPT-4 (Cui et al., 2023). We will make use of $\text{Err}_{\pi_{\text{ref}}}(\hat{r})$ to characterize the statistical rate attained by RDPO in the next section. Before delving into the proofs, let us state an important assumption that we leverage for all our technical results.

**Assumption 1.** *For any $\pi_{\text{ref}}$ and reward $r : \mathcal{X}\times\mathcal{A} \to [R_{\min}, R_{\max}]$, let $\pi_\beta^\star$ defined as in Equation (3). Then, we assume that we have access to a policy class $\Pi$ such that $\pi_\beta^\star \in \Pi$.*

The assumption says that the policy class is expressive enough to realize the closed form solution of Eq.1 corresponding to any bounded reward function. Moreover, we assume that each pair of candidate actions is sampled from the reference policy, i.e., $a_i \sim \pi_{\text{ref}}(\cdot|x_i)$ and $a_i' \sim \pi_{\text{ref}}(\cdot|x_i)$. We can now state our theoretical results starting from RDPO.

### 4.1 THEORETICAL GUARANTEES FOR RDPO: ACCELERATION AND ROBUSTNESS

In order to present the bounds more clearly, let us define $\text{Err}_{\text{DPO}}(N, \delta) := \frac{32 R_{\max}^2 e^{4R_{\max}} \log(|\Pi||\mathcal{R}|/\delta)}{N}$. The definition is due to the fact that DPO finds a policy $\pi_{\text{DPO}}$ enjoying the following guarantees proven, (see for example Huang et al. (2024, Theorem 3.1)),

$$\langle \pi^\star - \pi_{\text{DPO}}, r^\star \rangle \leq \mathcal{O}\left(\sqrt{\Delta D(\Pi)\text{Err}_{\text{DPO}}(N, \delta)}\right) \quad \text{w.p. } 1-\delta, \qquad (11)$$

---

[6]https://huggingface.co/datasets/HuggingFaceH4/ultrafeedback_binarized

where $C^\star = \sum_{x \in \mathcal{X}} \nu_0(x) \sum_{a \in \mathcal{A}} \pi^\star(a|x) \left| \frac{\pi^\star(a|x)}{\pi_{\text{ref}}(a|x)} \right|$ is the so called single policy concentrability coefficient and $\Delta D(\Pi) := \max_{\pi \in \Pi}(C^\star + C^\pi - 2D_{\text{KL}}(\pi, \pi_{\text{ref}}))/2$ . We now state our main result for RDPO.

**Theorem 4.1.** *Let $\pi_{\text{out}}$ be the policy with parameters minimizing $\mathcal{L}_{\text{RDPO}}$ given in equation RDPO with hyperparameters $\beta = \beta'/(1 - \alpha)$ and $\beta_1 = \beta'/\alpha$ where $\beta' = \sqrt{2/(3\Delta D(\Pi))} \left( 1/\text{Err}_{\text{DPO}}(N,\delta) + 1/\text{Err}_{\pi_{\text{ref}}}(\hat{r}) \right)^{-1}$ and $\alpha = (1 + \text{Err}_{\pi_{\text{ref}}}(\hat{r})/\text{Err}_{\text{DPO}}(N, \delta))^{-1}$ and constraints given by the policy class $\Pi' \subset \Pi$ such that $\pi \in \Pi' \implies \beta \Delta_{\log \pi/\pi_{\text{ref}}} - \frac{\beta}{\beta_1} \Delta_{\hat{r}} \in [-R_{\max}, R_{\max}]$. Then, it holds that with probability at least $1 - \delta$*

$$\langle \pi^\star - \pi_{\text{out}}, r^\star \rangle \leq \mathcal{O}\left( \sqrt{\Delta D(\Pi) \min\left( \text{Err}_{\text{DPO}}(N, \delta), \text{Err}_{\pi_{\text{ref}}}(\hat{r}) \right)} \right), \qquad (12)$$

*where $\Delta D(\Pi) := \max_{\pi \in \Pi}(C^\star + C^\pi - 2D_{\text{KL}}(\pi, \pi_{\text{ref}}))/2$.*

A stronger theoretical bound where $\Delta D(\Pi)$ is replaced by the smaller quantity $C^\star$ can be obtained with an additional $\chi^2$ regularization as shown in Theorem C.1 in Appendix C. However, we noticed that the $\chi^2$ can make the algorithm numerically unstable in practice. The rate proven in Eq. 12 features two important theoretical properties that we discuss next.

**Acceleration.** When the ratings $\hat{r}$ are accurate, it is expected that $\text{Err}_{\pi_{\text{ref}}}(\hat{r}) \leq \text{Err}_{\text{DPO}}(N, \delta)$. Therefore, the bound in Theorem 4.1 predicts that $\langle \pi^\star - \pi_{\text{out}}, r^\star \rangle \leq \mathcal{O}\left( \sqrt{\Delta D(\Pi)\text{Err}_{\pi_{\text{ref}}}(\hat{r})} \right)$, which is faster than the DPO rate in Eq. 11.

**Robustness.** Even if the ratings $\hat{r}$ are inaccurate, i.e., $\text{Err}_{\pi_{\text{ref}}}(\hat{r}) \geq \text{Err}_{\text{DPO}}(N, \delta)$, the guarantees for RDPO do not collapse. Indeed, Theorem 4.1 predicts $\langle \pi^\star - \pi_{\text{out}}, r^\star \rangle \leq \mathcal{O}\left( \sqrt{\Delta D(\Pi)\text{Err}_{\text{DPO}}(N, \delta)} \right)$, which is the same bound achieved by DPO-like algorithms.

Moreover, Theorem 4.1 provides important guidance for the choice of the hyperparameters.

**Theoretical guidance for the choice of $\beta, \beta_1$ in practice.** Using the definitions $\beta = \beta'/(1 - \alpha)$ and $\beta_1 = \beta'/\alpha$, we have that $\beta/\beta_1 = \frac{\alpha}{1-\alpha}$. At this point, plugging in the value of $\alpha$ given in Theorem 4.1, one obtains the relation $\beta/\beta_1 = \text{Err}_{\text{DPO}}(N, \delta)/\text{Err}_{\pi_{\text{ref}}}(\hat{r})$. Therefore, our analysis suggests to set the rating trust $1/\beta_1$ to the inverse of the regularization parameter, i.e. $1/\beta$, times the errors ratio $\text{Err}_{\text{DPO}}(N,\delta)/\text{Err}_{\pi_{\text{ref}}}(\hat{r})$. Therefore, in practice, *the more we trust the rating over the ranking, the larger $\beta/\beta_1$ should be.* As an example, $\beta$ can be set to the default value in the DPO implementation, e.g., $\beta = 0.1$, and $\beta_1$ can be chosen consequently to obtain the desired ratio $\beta/\beta_1$.

### 4.2 THEORETICAL GUARANTEES FOR ML-RDPO

We now prove theoretical results for ML-RDPO similar to those we proved for RDPO in Section 4.1.

**Theorem 4.2.** *Let us assume that we have conditional independence between $z_i$ and $\Delta_{\hat{r}}^i$ given $x_i, a_i, a_i'$, and rating gaps are Gaussian $\Delta_{\hat{r}}^i \sim \mathcal{N}(\Delta_{r^\star}^i, \mathbb{V})$, for all $i \in [N]$. Under these condition, let $\pi_{\text{out}}$ be a solution to Eq. ML-RDPO optimized over a policy class $\tilde{\Pi} \subset \Pi$ such that $\pi \in \tilde{\Pi}$ implies $\beta \Delta_{\log \pi/\pi_{\text{ref}}} \in [-R_{\max}, R_{\max}]$ with $\beta = \tilde{\mathcal{O}}\left( \sqrt{\frac{\min\{R_{\max}^2 e^{R_{\max}}, \mathbb{V} + R_{\max}^2\} \log(|\Pi|/\delta)}{\Delta D(\Pi)N}} \right)$. Then, with probability $1 - \delta$, $\langle \pi^\star - \pi_{\text{out}}, r^\star \rangle \leq \tilde{\mathcal{O}}\left( \sqrt{\frac{\Delta D(\Pi) \min\{e^{R_{\max}} R_{\max}^2, R_{\max}^2 + \mathbb{V}\} \log(|\Pi|/\delta)}{N}} \right)$.*

Notice also that Theorem 4.2 can be extended to general $f$-divergences rather than the KL divergence. In particular, as shown in Theorem C.2, with an additional $\chi^2$ term, we can obtain a tighter bound where $C^\star$ replaces $\Delta D(\Pi)$. We now highlight some interesting features of this bound.

**Acceleration and Robustness Properties.** As for RDPO, ML-RDPO enjoys both the acceleration and robustness properties. Regarding acceleration, we can see that when the variance $\mathbb{V}$ is low, i.e., $\mathbb{V} \leq \mathcal{O}\left( e^{R_{\max}} R_{\max}^2 \right)$, then Theorem 4.2 shows a better rate for ML-RDPO than the one for DPO given by Eq. 11. In particular, when the variance is small, the exponential dependence in $R_{\max}$

can be avoided. The only prior work that avoids the $e^{R_{\max}}$ dependence is Chen et al. (2025), but it requires online generation of new responses.

Note that the proof technique used in Theorem 4.2 can be used to prove a new bound for Distilled-DPO (Fisch et al., 2024) of order $\mathcal{O}\left(\sqrt{\Delta D(\Pi)(\mathbb{V} + R_{\max}^2)\log(|\Pi|/\delta)/N}\right)$. This allows us to conclude that ML-RDPO is no worse than the best between DPO and ML-RDPO with high probability.

**Comparison with RDPO.** As we previously mentioned, $\mathcal{L}_{\mathrm{ML-RDPO}}$ is written as a term that involves only the rating information and one that involves only the ranking dataset. This fact makes ML-RDPO easily generalizable to the setting where the rating feedback is observed for only certain responses in the dataset. On the contrary, RDPO can not be directly applied in this setting because $\mathcal{L}_{\mathrm{RDPO}}$ does not decompose into a rating and ranking term. We elaborate on this in Appendix B.2 and show the efficiency of ML-RDPO with partial rating information in practice in Figure 4. A downside for ML-RDPO is the Gaussian assumption on the rating, which is avoided by RDPO.

## 5 RELATED WORK ON LEARNING FROM RANKING AND RATING

We will compare our algorithms with three main families of alignment algorithms classified depending on the required input. In particular, we compared with algorithms which require only rankings, only ratings, and finally algorithms which work exactly in our setting where both sources of information are available. Next, we review the state-of-the-art algorithms in each of these families.

Alignment with only ranking information has been widely studied and led to popular algorithms such as DPO (Rafailov et al., 2023), which leverages the Bradley-Terry model, and IPO (Azar et al., 2024), which applies to general preference models. We will use both IPO and DPO as baselines. Moreover, we will compare with SIMPO Meng et al. (2024), which modifies DPO by removing the regularization towards $\pi_{\mathrm{ref}}$ and dividing the log ratios by the length of the responses in the dataset.

Among the ratings-only alignment algorithms, we compare with Distilled DPO (abbreviated as DDPO) (Fisch et al., 2024), which uses the squared loss to learn a policy whose log ratios difference is close to the observed ratings difference. Interestingly, our RIPO is quite similar to DDPO, albeit their derivations follow different perspectives.

Finally, alignment from rating and ranking information has been studied only more recently. For example, Adler et al. (2024a) introduced RPO which minimizes $\mathbb{KL}(\mathrm{Bern}(\sigma(\Delta_\theta^i)), \mathrm{Bern}(\sigma(\Delta_{\hat{r}}^i)))$. Unfortunately, RPO does not enjoy the performance guarantees attained by RDPO and ML-RDPO, and more importantly, it is difficult to tune due to the non-smoothness of its loss function. We also compare with another rating-augmented alignment algorithm dubbed MAPPO (Lan et al., 2025) that minimizes $\sum_{i=1}^{N} -\log\sigma\left(\beta\log\left(\pi_\theta(a_i^+|x_i)/\pi_{\mathrm{ref}}(a_i^+|x_i)\right) - \beta\Delta_{\hat{r}}^i\log\left(\pi_\theta(a_i^-|x_i)/\pi_{\mathrm{ref}}(a_i^-|x_i)\right)\right)$, that is the DPO loss up to the fact that the $\beta$ for the discarded response is multiplied by the rating gap. For completeness, we included an extended literature review in Appendix A.

## 6 EXPERIMENTS

In this section, we empirically evaluate our proposed algorithms, RDPO, RIPO, and ML-RDPO, and we demonstrate their solid performance in comparison to a number of DPO-style methods across a wide range of LLMs and evaluation benchmarks. The experiments verify our theoretical findings, especially in terms of robustness and faster convergence rate (acceleration).

**Experiments to assess acceleration.** Here we aim at verifying the favorable convergence properties of our algorithms shown in Theorem 4.1 and Theorem 4.2. We consider having access to high-quality rating information, $\hat{r}$, in the `ultrafeedback_binarized` dataset, and compare our methods to a diverse set of baselines: those that only use ranking information, DPO (Rafailov et al., 2023), IPO (Azar et al., 2024), and SIMPO (Meng et al., 2024), those that only use the rating information, DDPO(Fisch et al., 2024)), and finally those that leverage both pieces of information, RPO (Adler et al., 2024a) and MAPPO (Lan et al., 2025).

For most algorithms, we set the DPO hyperparameters to $\beta = 0.1$ and learning rate $10^{-6}$. For RDPO and ML-RDPO, we tune $\beta_1$ and $\mathbb{V}$ in Zephyr-7B as shown in Table 1 in Appendix F.4 and keep their

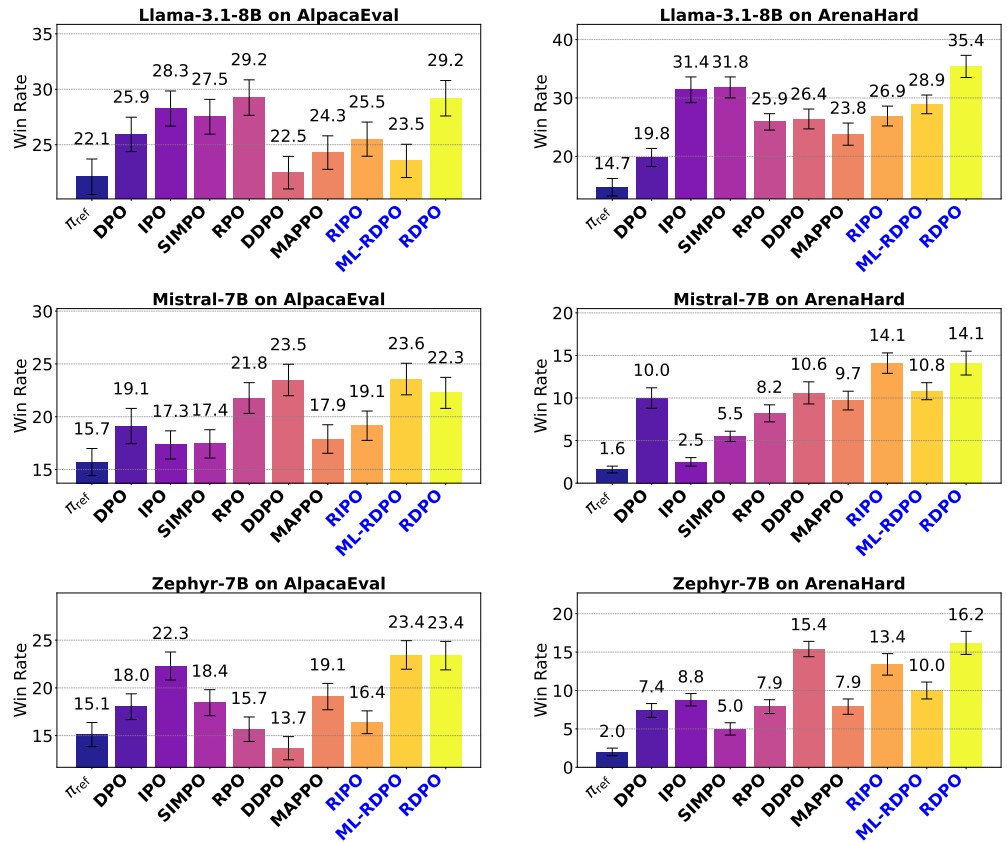

Figure 2: Win rates against GPT4, judged by Claude-Sonnet-3.5 v2, in AlpacaEval and ArenaHard.

best values, $\beta_1 = 1/10$ and $\mathbb{V} = 1/100$, for the experiments with Llama-3.1-8B and Mistral-7B models. For obtaining good results with DDPO, we reduced the learning rate to $10^{-7}$, while for IPO we used length normalization, larger $\beta$, i.e., $\beta = 0.8$, and a smaller learning rate $10^{-7}$.

In Figure 2, we start the fine-tuning from each of the specified base models (Llama, Mistral, Zephyr), and then evaluate the fine-tuned checkpoints in terms of their win-rates against GPT-4, judged by Claude-Sonnet-3.5 v2, over AlpacaEval and ArenaHard benchmarks. RDPO performs consistently well, being the best performing method in 3 out of the 6 experiments, and the second or third best in the others. ML-RDPO is less consistent, but it is the best performing method for Mistral-7B and Zephyr-7B in AlpacaEval. RPO is the best for Llama-3.1-8B in AlpacaEval, but the performance is sub-optimal in the other cases. The other methods that are only based on either rating or ranking information are less efficient in all experiments.

**Experiments to assess robustness.** Rating information is harder to collect and can be noisier than its ranking counterparts. Therefore, we evaluate the robustness of RDPO by corrupting the rating information in two different forms. In the first experiment, we swap the score of the chosen and discarded responses for a certain proportion (0%, 10%, 30%) of the training data. As can be seen in Figure 3a, when we use a small value $\beta_1 = 1/40$ (high trust in rating information), the performance of RDPO worsens fast as the number of swaps is increased. On the other hand, a larger $\beta_1 = 1/10$ makes the performance of RDPO robust and (almost) unaffected by the corruption.

We can draw a similar conclusion from our second experiment shown in Figure 3b, where we add zero-mean Gaussian noise with increasing variance to the rating information (UltraFeedback scores). Overall, these experiments confirm our theoretical observation that no matter how "bad" the rating information is, i.e., for any value of $\mathrm{Err}_{\pi_{\mathrm{ref}}}(\hat{r})$, there exists a choice of $\beta_1$ such that RDPO is guaranteed to perform at least as well as DPO.

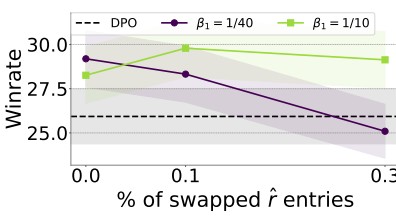 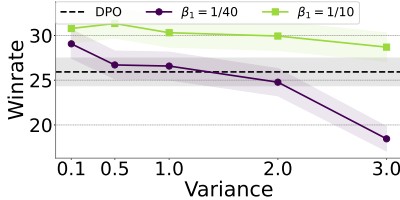

(a) Corruption via swaps, $\pi_{\mathrm{ref}} = $ `Llama-3.1-8B`.    (b) Corruption via noise, $\pi_{\mathrm{ref}} = $ `Llama-3.1-8B`.

Figure 3: Robustness to inaccurate ratings experiments

**Experiment with missing ratings.** Given that collecting rating information is more complicated than preference (ranking) data, it is reasonable to expect that in a practical scenario, some data points come only with ranking labels and without the rating gap information. We set up an experiment in Figure 4 to show the capability of ML-RDPO in handling training data with partial score labeling. In particular, we show that even when only $50\%$ of the training data contains rating information, ML-RDPO is preferable to DPO. Recall that, unlike RDPO, ML-RDPO does not need to have access to the rating gap for all the data-points in its training set. We provide the derivation of the ML-RDPO algorithm where some of the rating information is missing in Appendix B.2, and provide theoretical guarantees for this setting in Appendix D.

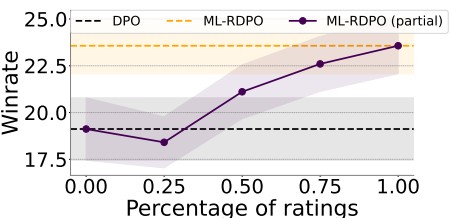

Figure 4: ML-RDPO on `ultrafeedback` with partial ratings observation, $\pi_{\mathrm{ref}} = $ `Mistral-7B`.

**Additional experiments.** We report additional experiments in Appendix F. In particular, in Appendix F.1, we report the pairwise win rates between any pair of considered algorithms at 7/8B scale. This serves as additional verification of the good performance of RDPO and ML-RDPO. In Appendix F.2, we repeat the corruption experiments on other choices of $\pi_{\mathrm{ref}}$, which confirms the trend observed in Figure 3b, especially using Mistral-7B as $\pi_{\mathrm{ref}}$. In Appendix F.3 we show that the gains from rating information are even more visible at smaller scales, using as $\pi_{\mathrm{ref}}$ the SmolLM2 models (Allal et al., 2025) of sizes 135M, 360M and 1.7B parameters. In particular, RDPO achieves an over $90\%$ win rate in all these cases. Finally, in Appendix F.5, we show that minimizing a modified RDPO loss, which encourages the restriction to the smaller policy class $\Pi'$ needed for the proof of Theorem 4.1, leads to improved results for Llama-3.1-8B and Mistral-7B as $\pi_{\mathrm{ref}}$. In particular, we add a piecewise linear penalty to encourage the implicit reward difference $\beta\Delta_\theta^i - \beta\Delta_{\hat{r}}^i/\beta_1$ to lie in the interval $[R_{\min} - R_{\max}, R_{\max} - R_{\min}]$.

## 7 CONCLUSIONS

We studied how DPO-style algorithms can efficiently leverage additional information in the form of *rating gap*, which informs the learner how much the preferred response is better than the rejected one. We presented three algorithms, RDPO, RIPO, and ML-RDPO, that have been derived using two different principled approaches. Our theoretical and empirical results show the better performance of our algorithms compared to methods that only use ranking information when the rating information is accurate. We also showed that our algorithms are robust to noise in the rating information, and can still perform well in this situation if their relevant hyper-parameters, $\beta_1$ and $\mathbb{V}$, are tuned properly.

A quick look at the losses of our proposed algorithms, we notice that they only leverage the rating gap, $\Delta_{\hat{r}}^i$, and not the individual ratings $\hat{r}(x_i, a_i^+)$ and $\hat{r}(x_i, a_i^-)$. This is positive because these algorithms can work with a weaker form of feedback (rating gap vs. individual rating), but as a consequence, they may not be able to use all the available information when the individual ratings are available. We leave developing principled algorithms, possibly inspired by our methods, that can make full use of the individual rating information as an interesting open question for future work.

## REPRODUCIBILITY STATEMENTS

All the hyperparameters used are stated in the text. In particular, Appendix F.4 reports a detailed ablation. We used the TRL library von Werra et al. (2020) as a starting codebase for our implementation.

## ETHICS STATEMENT

We do not foresee any ethical concern arising from our work.

## DECLARATION OF LLMS USAGE

We used LLMs as a helper in the implementation part. LLMs have not been used at all in deriving the theoretical results and in the writing of the manuscript.

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

## A EXTENDED RELATED WORKS DISCUSSION

This section outlines important related works that have been omitted from the main text due to space limitations.

**Learning from preferences and ratings**   The use of ratings estimated rewards has received little attention at the current stage, mainly because of the concern that the data collection becomes more complicated and inevitably noisier. However, there are some works that go against this belief and have shown practical benefit in using rating information. For example, Adler et al. (2024a), which introduced RPO and used it for the post training of a large (340B) language model after an initial post-training phase that was carried out from preferences only via DPO. In their case, the ratings allowed to use a large dataset with 300K examples with a rather permissive quality filtering on the chosen response. Their conclusion is therefore that bad quality chosen ( over even worse ones ) responses do not negatively affect the performance since they are accompanied by a low rate. Albeit RPO is effective in practice, as confirmed by our implementation, it does not enjoy the performance guarantees attained by RDPO and ML-RDPO, and it is difficult to tune due to the non-smoothness of its loss function.

RPO has also been generalized in Sun et al. (2025) to show that it can be seen as a general framework capturing various existing algorithms. Remaining on the algorithmic side, ratings information has been leveraged in deriving MAPPO Lan et al. (2025), which modifies the DPO objective using the rating gap to modify the regularization parameter $\beta$ in front of the log ratio of the loser response. As another example, Chen et al. (2024) provides a reward-based generalization of DPO dubbed NCA (Noisy Contrastive Alignment), which ensures that the probability of the winning response does not decrease, while this can (and in fact it does, very often) happen in DPO. However, it is questionable if such behavior is always desirable. For example, our intuition is that this should not happen if the rating of the winner's response is low. Moreover, notice that the same effect can be achieved by using an SFT term within an Online DPO scheme Pang et al. (2024). In this work, the authors assume rating feedback in order to assign the roles of winner and loser responses in the preference datasets generated iteratively by Online DPO without the need to resort to an LLM as a judge. Finally, rating information has been used within a data augmentation technique Zhang et al. (2024) to double the size of the preference dataset available for DPO training. Moreover, access to ratings has been used in the context of alignment from multi preferences, i.e. when multiple preferred and dispreferred responses are available in Gupta et al. (2024b; 2025). In passing, we notice that when only one preferred response and one rejected one is available MPO and W-MPO reduce to DPO. Interestingly, if the definition of $\Delta W_{\text{abs}}$ is replaced with $\Delta W(y) = S(y) - S_{\text{mean}}(x)$ and only two responses are available then W-MPO reduces to our Ratings DPO. In a similar vein, Gupta et al. (2024a) generalizes SIMPO Meng et al. (2024) to the case of multiple preferred and discared responses possibly with a weight associated with them for their version dubbed W-REFA.

**Learning from ratings only**   Several methods have focused on datasets given in the form of reward observations or margins ( reward differences) and developed methods to compute the soft optimal policy despite the computational hardness of the partition function. Examples are Distilled DPO Fisch et al. (2024) that studied in the context of offline direct preference optimization, the squared loss minimization suggested in Gao et al. (2024a;b) in the context of online RLHF for single or multi-turn, respectively. A similar approach is derived in Cai et al. (2023).

**Other variants of DPO with additional information**   Song et al. (2024) studied which kind of coverage conditions are necessary for DPO, IPO and in general purely offline alignment methods. Moreover, they relax the notion of coverage needed, resorting to online interaction with the environment but without invoking the preference oracle. This allows us to conclude that even without collecting new preferences, online access to the environment is beneficial for DPO. Similarly, Chang et al. (2024) studied preference optimization from the RLHF perspective when new trajectories ( from any desired initial prompt) can be collected and scored with the reward function, which maximizes the loglikelihood under the Bradley-Terry model.

Furthermore, Liu et al. (2024) tackled the likelihood displacement problem and mitigated the issue by adding to the algorithm an imitation learning component that increases the likelihood of generating the chosen responses. Cen et al. (2024) introduced a similar regularization mechanism by changing

the maximum likelihood estimation problem for the reward function. In particular, they look for the reward that maximizes the likelihood of the observed preferences and jointly makes the cumulative reward achievable by the best policy as high as possible. We think that it is an interesting open question to understand if the above methods can have provable benefits from ratings observations.

**Provably learning from preferences** Our work establishes clean theoretical guarantees under single policy concentrability only. It is therefore related to the line of literature started by Zhan et al. (2023) that studied the problem of offline alignment from a statistical learning viewpoint. The approaches in Zhan et al. (2023); Zhu et al. (2023) achieve indeed a single policy concentrability guarantee style. However, their algorithm is not implementable because it involves estimating confidence sets for the reward function and the transition dynamics in the multi-turn setting. Moreover, once the confidence sets are computed, they require solving a robust MDP problem Iyengar (2005); Wiesemann et al. (2013) using the confidence sets as uncertainty sets. Although some efficient policy gradient methods Li et al. (2023a); Kumar et al. (2025); Viano et al. (2021; 2022) and value-based methods Zouitine et al. (2024); Clavier et al. (2024) have been developed, this approach is considered not scalable for LLMs. Alternatively, Li et al. (2023c) implements pessimism, implementing pessimistic value iteration via elementwise bonuses similar to what is done in the optimistic setting with UCBVI Azar et al. (2017). While more elegant theoretically, this approach does not solve the scalability issue. A more practical pessimistic approach is obtained in Huang et al. (2024) via $\chi^2$ regularization, which also serves as a source of inspiration for our analysis.

**Learning from preferences imposing a margin** Our method is also related to all variants of DPO that imposes that the pseudoreward of the chosen completion is higher that the discarded completion one by at least a fixed margin. Examples are IPO that imposes the margin to be *exactly* $1/2$ via the squared loss, in the training of Llama2 Touvron et al. (2023) they use three possible discrete values for the margin in the log sigmoid loss. Notice that the log sigmoid does not penalize the overestimation of the margin so it encourages the pseudoreward difference to be at least as high as the margin but not to match the exact value. SimPO Meng et al. (2024) uses a fixed margin as in IPO but uses the log sigmoid loss instead of squares and neglects the effect of the reference policy in the definition of the pseudorewards. RDPO presented in this document can be seen as a generalization of the above approaches using a margin which is prompt completions dependent and takes values in a continuous set.

## B  GENERALIZATIONS OF RDPO AND ML-RDPO

In this appendix, we present two important extensions on RDPO and ML-RDPO. First, we present a generalization that allows to change the KL regularizer to any $f$-divergence. In particular, using as regularizer the sum of KL and $\chi^2$ divergence allows to prove statistical guarantees under single policy concentrability. Secondly, we show that we can handle disjoint datasets of rating and ranking. That is, we can apply our algorithm where the data are organized in two dataset: a classic ranking dataset $\mathcal{D}_{\mathrm{rank}} = \left\{ x_i, a_i^+, a_i^- \right\}_{i=1}^{N_{\mathrm{rank}}}$ and a rating only dataset $\mathcal{D}_{\mathrm{rat}} = \left\{ x_i', \tilde{a}_i, \bar{a}_i, \hat{r}(x_i', \tilde{a}_i) - \hat{r}(x_i', \bar{a}_i) \right\}_{i=1}^{N_{\mathrm{rat}}}$. We show that in this setting RDPO requires the intermediate step of estimating the unobserved ratings. That is, the rating gaps over the ranking dataset which are $\left\{ \hat{r}(x_i, a_i^+) - \hat{r}(x_i, a_i^-) \right\}$. Surprisingly, ML-RDPO avoids such step completely making it a more attractive alternative to handle this setting. We elaborate on this in Appendix B.2.

### B.1  DIFFERENT CHOICES FOR THE REGULARIZER IN EQ. 1

A natural generalization of RDPO can be derived replacing the KL regularization in Eq. 8 with other possible statistical divergences. An important case, is considering regularization given by the sum of KL and $\chi^2$ divergence which gives the following RLHF-like problem

$$\underset{\pi \in \Pi}{\mathrm{argmax}} \left\langle \pi, \frac{\hat{r}_{BT}(x,a)}{\beta} + \frac{\hat{r}(x,a)}{\beta_1} \right\rangle - D_{\mathrm{KL}}(\pi, \pi_{\mathrm{ref}}) - D_{\chi^2}(\pi, \pi_{\mathrm{ref}}) \tag{13}$$

---

**Algorithm 1** RDPO with arbitrary regularizer

---

1: **Input:** Datasets $\mathcal{D} = \left\{ (x_i, a_i, a_i', z_i, \Delta_{\hat{r}}^i) \right\}$, $\beta'$, $\alpha$, Policy class $\Pi'$, convex increasing function $\phi : \mathbb{R} \to \mathbb{R}$.

2: Set $\beta = \frac{\beta'}{1-\alpha}$, $\beta_1 = \frac{\beta'}{\alpha}$.

3: Let $\Delta_{\phi\pi/\pi_{\mathrm{ref}}}(x, a, b) := \beta\phi\left(\frac{\pi(a|x)}{\pi_{\mathrm{ref}}(a|x)}\right) - \beta\phi\left(\frac{\pi(b|x)}{\pi_{\mathrm{ref}}(b|x)}\right)$

4: **Return:** $\pi_{\mathrm{out}} \in \mathrm{argmin}_{\pi\in\Pi'} \sum_{i=1}^{N} -\log\sigma\left(\beta\Delta_{\phi\pi/\pi_{\mathrm{ref}}}(x_i, a_i, a_i') - \frac{\beta}{\beta_1}\Delta_{\hat{r}}^i\right)$.

---

where $\hat{r}_{BT} \in \mathrm{argmax}_{r\in\mathcal{R}_\Pi} \sum_{i=1}^{N} \log\sigma(r(x_i, a_i^+) - r(x_i, a_i^-))$ is the Bradley-Terry reward estimator and $D_{\mathrm{KL}}(\pi, \pi_{\mathrm{ref}})$, $D_{\chi^2}(\pi, \pi_{\mathrm{ref}})$ are defined as follows

$$D_{\mathrm{KL}}(\pi, \pi') = \sum_{x\in\mathcal{X}} \nu_0(x) \sum_{a\in\mathcal{A}} \pi(a|x) \log\frac{\pi(a|x)}{\pi'(a|x)}$$

and of the $\chi^2$-squared divergence defined as

$$D_{\chi^2}(\pi, \pi') = \frac{1}{2}\left(\sum_{x\in\mathcal{X}} \nu_0(x) \sum_{a\in\mathcal{A}} \pi(a|x)\frac{\pi(a|x)}{\pi'(a|x)} - 1\right).$$

It is easy to see that since $2x - 1 \geq \log(x)$ holds for any $x > 0$, it also holds that for a fixed policy pair $\pi, \pi'$, we have that $D_{\chi^2}(\pi, \pi) \geq D_{\mathrm{KL}}(\pi, \pi')$. However, the KL term is still important to ensure that Eq. 13 admits the following closed form solution

$$\pi_{\mathrm{out}}(a|x) \propto \pi_{\mathrm{ref}}(a|x)\phi_{\mathrm{KL}+\chi^2}^{-1}\left(\frac{\hat{r}_{BT}(x, a)}{\beta} + \frac{\hat{r}(x, a)}{\beta_1} - \zeta(x)\right).$$

where we used $\phi_{\mathrm{KL}+\chi^2}(x) := \log(x) + x$. The next Lemma shows that, when the reward class is parametrized by the policy parameters, the computation of $\pi_{\mathrm{out}}$ can be carried out without an explicit estimation of $\hat{r}_{BT}$.

**Lemma B.1.** *Generalization of RDPO. Let us consider an increasing invertible function $\phi : \mathbb{R}^+ \to \mathbb{R}$ and, given an arbitrary policy class $\Pi$, let us define the reward class $\mathcal{R}_\Pi$ defined as*

$$\mathcal{R}_\Pi = \left\{ r : \quad s.t. \quad \exists\pi \in \Pi \quad r(x, a) = \beta\phi\left(\frac{\pi(a|x)}{\pi_{\mathrm{ref}}(a|x)}\right) - \frac{\beta}{\beta_1}\hat{r}(x, a) + \beta\zeta(x) \right\}, \qquad (14)$$

*and the Bradley-Terry reward estimate $\hat{r}_{BT}$ computed as*

$$\hat{r}_{BT} = \mathrm{argmax}_{r\in\mathcal{R}_\Pi} \sum_{i=1}^{N} \log\sigma(r(x_i, a_i^+) - r(x_i, a_i^-)).$$

*Moreover, let us consider $\pi_{\mathrm{out}}$ defined as*

$$\pi_{\mathrm{out}}(a|x) \propto \pi_{\mathrm{ref}}(a|x)\phi^{-1}\left(\frac{\hat{r}_{BT}(x, a)}{\beta} + \frac{\hat{r}(x, a)}{\beta_1} - \zeta(x)\right).$$

*Then, $\pi_{\mathrm{out}}$ is in the solution set of the following optimization problem, which corresponds to a generalization of $\mathcal{L}_{RDPO}$. In formulas, we have*

$$\pi_{\mathrm{out}} \in \mathrm{argmin}_{\pi\in\Pi} \sum_{i=1}^{N} -\log\sigma\left(\beta\Delta_{\phi\pi/\pi_{\mathrm{ref}}}^i - \frac{\beta}{\beta_1}\Delta_{\hat{r}}^i\right), \qquad (15)$$

*where $\Delta_{\phi\pi/\pi_{\mathrm{ref}}}^i = \Delta_{\phi\pi/\pi_{\mathrm{ref}}}(x_i, a_i, b_i) = \beta\phi\left(\frac{\pi(a_i|x_i)}{\pi_{\mathrm{ref}}(a_i|x_i)}\right) - \beta\phi\left(\frac{\pi(b_i|x_i)}{\pi_{\mathrm{ref}}(b_i|x_i)}\right)$.*

For $\phi(x) = \log(x)$, we recover the original DPO loss. For $\phi(x) = \phi_{\mathrm{KL}+\chi^2}(x) := x + \log(x)$ we obtain an RDPO version with an additional $\chi^2$ regularization term, which has been shown useful in Huang et al. (2024) to recover better theoretical results. Moreover, this generalization allows for rating gaps informed versions of $f$-DPO Wang et al. (2023).

*Proof.* Let us consider the log likelihood maximization of the ranking under the Bradley-Terry model ( see Eq. 2) and let us choose as reward class $\mathcal{R}$ the following class

$$\mathcal{R}_\Pi = \left\{ r : \text{ s.t. } \exists \pi \in \Pi \quad r(x, a) = \beta\phi\left(\frac{\pi(a|x)}{\pi_{\text{ref}}(a|x)}\right) - \frac{\beta}{\beta_1}\hat{r}(x, a) + \beta\zeta(x) \right\}.$$

We can perform a change of variable to obtain the following equality,

$$\min_{r \in \mathcal{R}_\Pi} \sum_{i=1}^{N} -\log \sigma \left( r(x_i, a_i^+) - r(x_i, a_i^-) \right)$$

$$= \min_{\pi \in \Pi} - \sum_{i=1}^{N} \log \sigma \left( \beta\phi\left(\frac{\pi(a_i^+|x_i)}{\pi_{\text{ref}}(a_i^+|x_i)}\right) - \beta\phi\left(\frac{\pi(a_i^-|x_i)}{\pi_{\text{ref}}(a_i^-|x_i)}\right) - \frac{\beta}{\beta_1}\left(\hat{r}(x_i, a_i^+) - \hat{r}(x_i, a_i^-)\right) \right.$$

$$\left. - \beta\zeta(x) + \beta\zeta(x) \right)$$

$$= \min_{\pi \in \Pi} \sum_{i=1}^{N} -\log \sigma \left( \beta\phi\left(\frac{\pi(a_i^+|x_i)}{\pi_{\text{ref}}(a_i^+|x_i)}\right) - \beta\phi\left(\frac{\pi(a_i^-|x_i)}{\pi_{\text{ref}}(a_i^-|x_i)}\right) - \frac{\beta}{\beta_1}\left(\hat{r}(x_i, a_i^+) - \hat{r}(x_i, a_i^-)\right) \right)$$

$$= \min_{\pi \in \Pi} \sum_{i=1}^{N} -\log \sigma \left( \beta\Delta_{\phi\pi/\pi_{\text{ref}}}^i - \frac{\beta}{\beta_1}\Delta_{\hat{r}}^i \right),$$

which is the loss we report in Algorithm 1. $\square$

Moreover, we have the following generalization of ML-RDPO.

**Generalization of ML-RDPO**   Using the reward class $\mathcal{R}_\Pi$ for a general $\phi$ given in Equation (14) as constraint for the following optimization problem $\tilde{r} = \arg\max_{r \in \mathcal{R}_\Pi} \log P_r^{\text{rat,rank}}(\mathcal{D})$ and using the conditional independence assumption and Gaussianity of the ratings gap as in the ML-RDPO derivation, as done in the proof of Lemma 3.1, we arrive at the following minimization problem to compute the output policy,

$$\pi_{\text{out}} \in \arg\min_{\pi \in \Pi} \sum_{i=1}^{N} -\log \sigma(\Delta_{\phi\pi/\pi_{\text{ref}}}^i) + (2\mathbb{V})^{-1}\left(\Delta_{\hat{r}}^i - \Delta_{\phi\pi/\pi_{\text{ref}}}^i\right)^2. \tag{16}$$

Again, the only difference is that $\Delta_{\phi\pi/\pi_{\text{ref}}}^i$ replaces the log gaps $\Delta_{\log \pi/\pi_{\text{ref}}}^i$.

## B.2   EXTENSION TO HETEROGENEOUS DATASETS AND MISSING RATINGS

In this section, we explain how RDPO and ML-RDPO can be extended to the case where the rating and ranking datasets are only partly overlapping or even disjoint. As previously mentioned, we denote the ranking dataset as $\mathcal{D}_{\text{rank}} = \left\{x_i, a_i^+, a_i^-\right\}_{i=1}^{N_{\text{rank}}}$ and the rating only dataset $\mathcal{D}_{\text{rat}} = \left\{x_i', \tilde{a}_i, \bar{a}_i, \hat{r}(x_i', \tilde{a}_i) - \hat{r}(x_i', \bar{a}_i)\right\}_{i=1}^{N_{\text{rat}}}$. An important special case is represented by the setting where the ranking is available for all data pairs in the data set, but the rating is present only on a portion of it. Such a situation is quite likely to happen since rating gaps are more costly to obtain.

**RDPO for heterogeneous data**   In order to apply RDPO to this setting we need to maintain a rating estimator $\hat{r}_{\text{rat}}$ computed via least square regression using the available observations $\{\hat{r}(x_i', \tilde{a}_i) - \hat{r}(x_i', \bar{a}_i)\}_{i=1}^{N_{\text{rat}}}$ as regression targets. In particular, as a first step, we compute,

$$\hat{r}_{\text{rat}} = \arg\min_{r \in \mathcal{R}} \sum_{i=1}^{N_{\text{rat}}} (r(x_i', \tilde{a}_i) - r(x_i', \bar{a}_i) - \hat{r}(x_i', \tilde{a}_i) + \hat{r}(x_i', \bar{a}_i))^2$$

At this point, we can use RDPO to evaluate the reward model on the ranking dataset. This gives Algorithm 2.

---

**Algorithm 2** RDPO for heterogeneous data

---

1: **Inputs:**

- $\phi(x) = \gamma x + \log(x)$, $\beta$, $\alpha$.
- Policy class $\Pi'$ such that $\pi \in \Pi'$ implies $\left| \beta \Delta_{\phi\pi/\pi'_{\mathrm{ref}}} \right| \leq R_{\max} - R_{\min}$,
- Datasets:
  - $\mathcal{D}_{\mathrm{rank}} = \left\{ x_i, a_i^+, a_i^- \right\}_{i=1}^{N_{\mathrm{rank}}}$
  - $\mathcal{D}_{\mathrm{rat}} = \{ x_i', \tilde{a}_i, \bar{a}_i, \hat{r}(x_i', \tilde{a}_i) - \hat{r}(x_i', \bar{a}_i) \}_{i=1}^{N_{\mathrm{rat}}}$

2: Set $\beta = \frac{\beta'}{1-\alpha}$ and $\beta_1 = \frac{\beta'}{\alpha}$.

3: Let us denote

- $\Delta_{\phi\pi/\pi_{\mathrm{ref}}}(x, a, b) = \beta\phi\left( \frac{\pi(a|x)}{\pi_{\mathrm{ref}}(a|x)} \right) - \beta\phi\left( \frac{\pi(b|x)}{\pi_{\mathrm{ref}}(b|x)} \right)$
- $\Delta_{\phi\pi/\pi_{\mathrm{ref}}}^i = \Delta_{\phi\pi/\pi_{\mathrm{ref}}}(x_i, a_i^+, a_i^-)$

4: Compute the reward estimator based on ratings

$$\hat{r}_{\mathrm{rat}} = \operatorname*{argmin}_{r \in \mathcal{R}} \sum_{i=1}^{N_{\mathrm{rat}}} (r(x_i', \tilde{a}_i) - r(x_i', \bar{a}_i) - \hat{r}(x_i', \tilde{a}_i) + \hat{r}(x_i', \bar{a}_i))^2$$

5: Set $\Delta_{\hat{r}_{\mathrm{rat}}}^i = \hat{r}_{\mathrm{rat}}(x_i, a_i^+) - \hat{r}_{\mathrm{rat}}(x_i, a_i^-)$ for all $i \in [N_{\mathrm{rank}}]$

6: **Return:** $\pi_{\mathrm{out}}$ such that $\pi_{\mathrm{out}} \in \operatorname{argmax}_{\pi \in \Pi'} \sum_{i=1}^{N_{\mathrm{rank}}} \log \sigma\left( \beta \Delta_{\phi\pi/\pi_{\mathrm{ref}}}^i - \frac{\beta}{\beta_1} \Delta_{\hat{r}_{\mathrm{rat}}}^i \right)$.

---

**ML-RDPO for heterogeneous data**  ML-RDPO is particularly convenient in the heterogeneous case because it avoids the least square regression problem for computing $\hat{r}_{\mathrm{rat}}$. For this derivation, we assume that the datasets $\mathcal{D}_{\mathrm{rank}}$ and $\mathcal{D}_{\mathrm{rat}}$ are independent. Therefore $P_r^{\mathrm{rat,rank}}(\mathcal{D}_{\mathrm{rat}}, \mathcal{D}_{\mathrm{rank}}) = P_r^{\mathrm{rat}}(\mathcal{D}_{\mathrm{rat}}) P_r^{\mathrm{rank}}(\mathcal{D}_{\mathrm{rank}})$. Under this condition, we can rewrite the joint maximum likelihood problem as follows for any reward class $\mathcal{R}$

$$\bar{r} = \operatorname*{argmax}_{r \in \mathcal{R}} \log \left( P_r^{\mathrm{rank}}(\mathcal{D}_{\mathrm{rank}}) P_r^{\mathrm{rat}}(\mathcal{D}_{\mathrm{rat}}) \right)$$

$$= \operatorname*{argmax}_{r \in \mathcal{R}} \log \left( P_r^{\mathrm{rank}} \left( \left\{ x_i, a_i^+, a_i^- \right\}_{i=1}^{N_{\mathrm{rank}}} \right) P_r^{\mathrm{rat}} \left( \left\{ x_i', \tilde{a}_i, \bar{a}_i, \Delta_{\hat{r}}^i \right\}_{i=1}^{N_{\mathrm{rat}}} \right) \right)$$

$$= \operatorname*{argmax}_{r \in \mathcal{R}} \sum_{i=1}^{N_{\mathrm{rank}}} \log P_r^{\mathrm{rank}} \left( z_i | x_i, a_i, a_i' \right) + \sum_{i=1}^{N_{\mathrm{rat}}} \log P_r^{\mathrm{rat}} \left( \Delta_{\hat{r}}^i | x_i', \tilde{a}_i, \bar{a}_i \right)$$

Now, plugging the Bradley-Terry model with reward $r \in \mathcal{R}$ as probability law $P_r^{\mathrm{rank}}$ and the Gaussian density of variance $\mathbb{V}$ and mean $\Delta_r^i := r(x_i', \tilde{a}_i) - r(x_i', \bar{a}_i)$ for $P_r^{\mathrm{rat}}$, we obtain the following loss

$$\operatorname*{argmin}_{r \in \mathcal{R}} \sum_{i=1}^{N_{\mathrm{rank}}} -\log \sigma(r(x_i, a_i^+) - r(x_i, a_i^-)) + \sum_{i=1}^{N_{\mathrm{rat}}} \left( \Delta_{\hat{r}}^i - r(x_i', \tilde{a}_i) + r(x_i, \bar{a}_i) \right)^2.$$

Then, choosing the reward class

$$\mathcal{R} = \left\{ r \mid \exists \pi \in \tilde{\Pi} \text{ s.t. } r(x, a) = \beta\phi(\pi(a|x)/\pi_{\mathrm{ref}}(a|x)) \right\},$$

we can perform the change of variable from $r$ to $\beta\phi(\pi/\pi_{\mathrm{ref}})$

$$\operatorname*{argmin}_{\pi \in \tilde{\Pi}} \sum_{i=1}^{N_{\mathrm{rank}}} -\log \sigma(\Delta_{\phi\pi/\pi_{\mathrm{ref}}}(x_i, a_i^+, a_i^-)) + \sum_{i=1}^{N_{\mathrm{rat}}} \left( \Delta_{\hat{r}}^i - \Delta_{\phi\pi/\pi_{\mathrm{ref}}}(x_i, \tilde{a}_i, \bar{a}_i) \right)^2 \quad (17)$$

which is the ML-RDPO loss but with the sums taken on different set of data.

## C  OMITTED PROOFS

This section contains the technical proofs of all the results included in the main text.

We will make use of the following quantities

$$C^\pi = \sum_{x\in\mathcal{X}} \nu_0(x) \sum_{a\in\mathcal{A}} \pi(a|x) \frac{\pi(a|x)}{\pi_{\text{ref}}(a|x)} \quad C^\star = \sum_{x\in\mathcal{X}} \nu_0(x) \sum_{a\in\mathcal{A}} \pi^\star(a|x) \frac{\pi^\star(a|x)}{\pi_{\text{ref}}(a|x)}$$

Given the above definitions, notice that we have the following relations

$$\frac{1}{2}(C^\pi - 1) = D_{\chi^2}(\pi, \pi_{\text{ref}}) \geq D_{\text{KL}}(\pi, \pi_{\text{ref}})$$

for all policies $\pi \in \Pi_{\text{all}}$. Therefore, we also have that

$$\frac{1}{2}(C^\star - 1) = D_{\chi^2}(\pi^\star, \pi_{\text{ref}}) \geq D_{\text{KL}}(\pi^\star, \pi_{\text{ref}}).$$

Next, we prove the theoretical results in the order they appear in the main text.

## C.1 Proof of Lemma 3.1

*Proof.* The key idea of the proof is to use the conditional independence and Gaussian assumption to rewrite the joint likelihood in a simpler, computationally attractive fashion.

$$\tilde{r} = \underset{r\in\mathcal{R}}{\arg\max} \sum_{i=1}^N \log \mathbb{P}(x_i, a_i, a_i', \Delta_{\hat{r}}^i, z_i)$$

$$= \underset{r\in\mathcal{R}}{\arg\max} \sum_{i=1}^N \log P_r^{\text{rat,rank}}(\Delta_{\hat{r}}^i, z_i | x_i, a_i, a_i') \mathbb{P}(x_i, a_i, a_i')$$

$$= \underset{r\in\mathcal{R}}{\arg\max} \sum_{i=1}^N \log P_r^{\text{rat,rank}}(\Delta_{\hat{r}}^i, z_i | x_i, a_i, a_i') \quad \text{(Using conditional independence)}$$

$$= \underset{r\in\mathcal{R}}{\arg\max} \sum_{i=1}^N \log P_r^{\text{rank}}(z_i | x_i, a_i, a_i') + \sum_{i=1}^N \log P_r^{\text{rat}}(\Delta_{\hat{r}}^i | x_i, a_i, a_i')$$

At this point, let us use the assumption that $\Delta_{\hat{r}}^i | x_i, a_i, a_i'$ is a gaussian random variable of variance $\mathbb{V}$ and mean $r(x_i', a_i') - r(x_i', \tilde{a}_i)$, That is, we set $P_r^{\text{rat}}(\Delta_{\hat{r}}^i | x, a, b) = \frac{1}{\sqrt{2\pi\mathbb{V}}} \exp\left(-\frac{(\Delta_{\hat{r}}^i - \Delta_r^i)^2}{2\mathbb{V}}\right)$. Then, using the assumption that $P_r^{\text{rank}}$ follows the Bradley-Terry model. All in all, we arrive at the following maximization problem

$$\tilde{r} = \underset{r\in\mathcal{R}}{\arg\max} \sum_{i=1}^N \log \sigma(r(x_i, a_i^+) - r(x_i, a_i^-)) - \frac{\sum_{i=1}^N \left(r(x_i, a_i) - r(x_i, a_i') - \Delta_{\hat{r}}^i\right)^2}{2\mathbb{V}}.$$

Replacing the definition $\Delta_r^i = r(x_i, a_i) - r(x_i, a_i')$ concludes the proof. $\square$

## C.2 Proof of Theorem 4.1

We prove Theorem 4.1 in the following more general form. For the statement, we will use the following definition $\Delta_{\phi\pi/\pi_{\text{ref}}}(x, a, a') := \phi\left(\frac{\pi(a|x)}{\pi_{\text{ref}}(a|x)}\right) - \phi\left(\frac{\pi(a'|x)}{\pi_{\text{ref}}(a'|x)}\right)$.

**Theorem C.1.** *Let $\pi_{\text{out}}$ be the output of Algorithm 1 ran with $\phi(x) = \phi_{KL+\chi^2}(x) := x + \log(x)$, $\beta' = \sqrt{2/3C^\star} \left(1/\text{Err}_{\text{DPO}}(N,\delta) + 1/\text{Err}_{\pi_{\text{ref}}}(\hat{r})\right)^{-1}$, $\alpha = (1 + \text{Err}_{\pi_{\text{ref}}}(\hat{r})/\text{Err}_{\text{DPO}}(N,\delta))^{-1}$, and policy class $\Pi' \subset \Pi$ such that $\pi \in \Pi' \implies \beta\Delta_{\phi\pi/\pi_{\text{ref}}} - \frac{\beta}{\beta_1}\Delta_{\hat{r}} \in [-R_{\max}, R_{\max}]$. Then, it holds that*

$$\langle \pi^\star - \pi_{\text{out}}, r^\star \rangle \leq \mathcal{O}\left(\sqrt{C^\star \min\left(\text{Err}_{\text{DPO}}(N,\delta), \text{Err}_{\pi_{\text{ref}}}(\hat{r})\right)}\right), \quad w.p. \quad 1 - \delta.$$

*Moreover, if $\phi(x) = \log(x)$, Then, it holds that*

$$\langle \pi^\star - \pi_{\text{out}}, r^\star \rangle \leq \mathcal{O}\left(\sqrt{\Delta D(\Pi') \min\left(\text{Err}_{\text{DPO}}(N,\delta), \text{Err}_{\pi_{\text{ref}}}(\hat{r})\right)}\right), \quad w.p. \quad 1 - \delta.$$

*where $\Delta D(\Pi') := \max_{\pi\in\Pi'}(C^\star + C^\pi - 2D_{\text{KL}}(\pi, \pi_{\text{ref}}))/2$.*

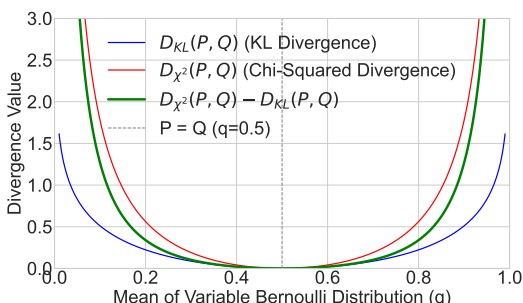

Figure 5: Comparison between $D_{\mathrm{KL}}$ and $D_{\chi^2}$ and their differences for Bernoulli random variables.

In order to prove the statements for $\phi(x) = \log(x)$ and $\phi(x) = \phi_{\mathrm{KL}+\chi^2}(x)$ in a unified manner, we consider the function $\phi(x) = \gamma x + \log(x)$ which recovers $\phi(x) = \log(x)$ for $\gamma = 0$ and $\phi(x) = \phi_{\mathrm{KL}+\chi^2}(x)$ for $\gamma = 1$. The proof of Theorem C.1 is divided in the next two lemmas: Lemma C.1 converts the suboptimality of the extracted policy $\pi_{\mathrm{out}}$ into the excess risk of the reward estimator corresponding to $\pi_{\mathrm{out}}$ defined as follows

$$\mathrm{Err}_{\pi_{\mathrm{ref}}}(\pi_{\mathrm{out}}) = \sum_{x \in \mathcal{X}} \nu_0(x) \sum_{a \in \mathcal{A}} \pi_{\mathrm{ref}}(a|x)\pi_{\mathrm{ref}}(b|x) \left( \Delta_{r^\star}(x, a, b) - \beta' \Delta_{\phi \pi_{\mathrm{out}}/\pi_{\mathrm{ref}}}(x, a, b) \right)^2.$$

Then, Lemma C.2 bounds $\mathrm{Err}_{\pi_{\mathrm{ref}}}(\pi_{\mathrm{out}})$ in high probability. Once these core facts are proven, the final result is simple to get, plugging in the upper bound given in Lemma C.2 into Lemma C.1.

**Lemma C.1.** *Let $\mathrm{Err}_{\max} \in \mathbb{R}$ be such that $\mathrm{Err}_{\pi_{\mathrm{ref}}}(\pi_{\mathrm{out}}) \le \mathrm{Err}_{\max}$ and let us run Algorithm 1 with $\gamma = 1$ and $\beta' = \sqrt{\frac{2\mathrm{Err}_{\max}}{3C^\star}}$, and policy class $\Pi'$. Then, it holds that for any comparator policy $\pi^\star \in \Pi$ it holds that*

$$\langle \pi^\star, r^\star \rangle - \langle \pi_{\mathrm{out}}, r^\star \rangle \le 8\sqrt{\mathrm{Err}_{\max} C^\star}.$$

*Moreover, setting $\gamma = 0$, $C^\star$ is replaced by $\Delta D(\Pi')$.*

An important remark is that the KL regularization is strictly suboptimal compared to $\chi^2$ regularization in the worst case, since we have that $\Delta D(\Pi') > C^\star$ unless we are in the spurious case of $\Pi' = \{\pi_{\mathrm{ref}}\}$. However, when the policy class $\Pi'$ does not allow large deviations from $\pi_{\mathrm{ref}}$ the difference between $D_{\chi^2}$ and $D_{\mathrm{KL}}$ is well controlled as shown in the Figure 5. Therefore, even with $\gamma = 0$ we expect good results in practice if the algorithm output remains close to $\pi_{\mathrm{ref}}$. Next, we state the proof of Lemma C.1.

*Proof.* Let us consider the notation

$$r_{\mathrm{out}}(x, a) = \frac{\beta'}{1 - \alpha} \phi \left( \frac{\pi_{\mathrm{out}}(a|x)}{\pi_{\mathrm{ref}}(a|x)} \right) - \frac{\alpha}{1 - \alpha} \hat{r}(x, a).$$

Rearranging the above definition, we obtain the following elementwise equality

$$(1 - \alpha)r_{\mathrm{out}} + \alpha\hat{r} = \beta' \phi \left( \frac{\pi_{\mathrm{out}}}{\pi_{\mathrm{ref}}} \right)$$

By Lemma E.1, we have that $\pi_{\mathrm{out}}$ is the solution of the following RLHF like problem

$$\pi_{\mathrm{out}} = \underset{\pi \in \Pi}{\mathrm{argmax}} \langle \pi, (1 - \alpha)r_{\mathrm{out}} + \alpha\hat{r} \rangle - \beta'\gamma D_{\chi^2}(\pi, \pi_{\mathrm{ref}}) - \beta' D_{\mathrm{KL}}(\pi, \pi_{\mathrm{ref}}) \qquad (18)$$

At this point, let $\bar{r} = (1 - \alpha)r_{\mathrm{out}} + \alpha\hat{r}$ and $D_F(\pi, \pi_{\mathrm{ref}}) := \gamma D_{\chi^2}(\pi, \pi_{\mathrm{ref}}) + D_{\mathrm{KL}}(\pi, \pi_{\mathrm{ref}})$. It holds that

$$\langle \pi^\star, \bar{r} \rangle - \beta' D_F(\pi^\star, \pi_{\mathrm{ref}}) \le \langle \pi_{\mathrm{out}}, \bar{r} \rangle - \beta' D_F(\pi_{\mathrm{out}}, \pi_{\mathrm{ref}})$$

We can now use the above fact to bound the suboptimality against a comparator $\pi^\star$

$$\langle \pi^\star, r^\star \rangle - \langle \pi_{\mathrm{out}}, r^\star \rangle =$$

$$\leq \langle \pi^\star, r^\star \rangle - \langle \pi^\star, \bar{r} \rangle + \beta' D_F(\pi^\star, \pi_{\mathrm{ref}}) + \langle \pi_{\mathrm{out}}, \bar{r} \rangle - \beta' D_F(\pi_{\mathrm{out}}, \pi_{\mathrm{ref}}) - \langle \pi_{\mathrm{out}}, r^\star \rangle$$

$$= \langle \pi^\star, r^\star \rangle - \langle \pi^\star, \bar{r} \rangle + \frac{\beta'\gamma}{2} C^\star + \beta' D_{\mathrm{KL}}(\pi^\star, \pi_{\mathrm{ref}})$$

$$+ \langle \pi_{\mathrm{out}}, \bar{r} \rangle - \frac{\beta'\gamma}{2} C^{\pi_{\mathrm{out}}} - \beta' D_{\mathrm{KL}}(\pi_{\mathrm{out}}, \pi_{\mathrm{ref}}) - \langle \pi_{\mathrm{out}}, r^\star \rangle. \tag{19}$$

In the final equality, we used the following chain of inequalities

$$D_F(\pi^\star, \pi_{\mathrm{ref}}) - D_F(\pi_{\mathrm{out}}, \pi_{\mathrm{ref}}) = \gamma D_{\chi^2}(\pi^\star, \pi_{\mathrm{ref}}) + D_{\mathrm{KL}}(\pi^\star, \pi_{\mathrm{ref}}) - \gamma D_{\chi^2}(\pi, \pi_{\mathrm{ref}})$$

$$- D_{\mathrm{KL}}(\pi, \pi_{\mathrm{ref}})$$

$$= \gamma \left( \frac{C^\star}{2} - 1 \right) + D_{\mathrm{KL}}(\pi^\star, \pi_{\mathrm{ref}}) - \gamma \left( \frac{C^\pi}{2} - 1 \right)$$

$$- D_{\mathrm{KL}}(\pi, \pi_{\mathrm{ref}})$$

$$= \gamma \frac{C^\star}{2} + D_{\mathrm{KL}}(\pi^\star, \pi_{\mathrm{ref}}) - \gamma \frac{C^\pi}{2} - D_{\mathrm{KL}}(\pi, \pi_{\mathrm{ref}})$$

At this point, we bound the reward estimation error under $\pi^\star$ as follows

$$\langle \pi^\star, r^\star - \bar{r} \rangle = \sum_{x \in \mathcal{X}} \nu_0(x) \sum_{a \in \mathcal{A}} \pi^\star(a|x)(r(x,a) - \bar{r}(x,a))$$

$$= \sum_{x \in \mathcal{X}} \nu_0(x) \sum_{a \in \mathcal{A}} \pi^\star(a|x)\pi_{\mathrm{ref}}(b|x)(r^\star(x,a) - r^\star(x,b) - \bar{r}(x,a) + \bar{r}(x,b))$$

$$- \underbrace{\sum_{x \in \mathcal{X}} \nu_0(x) \sum_{b \in \mathcal{A}} \pi_{\mathrm{ref}}(b|x)(\bar{r}(x,b) - r(x,b))}_{=\mathrm{Shift}} \tag{20}$$

At this point, we can perform the following change of measure trick. For notation compactness, let $f(x,a) = \sum_{b \in \mathcal{A}} \pi_{\mathrm{ref}}(b|x)(r^\star(x,a) - r^\star(x,b) - \bar{r}(x,a) + \bar{r}(x,b))$ and let $\pi$ be an arbitrary policy

$$\sum_{x \in \mathcal{X}} \nu_0(x) \sum_{a \in \mathcal{A}} \pi(a|x)f(x,a) = \sum_{x \in \mathcal{X}} \nu_0(x) \sum_{a \in \mathcal{A}} \frac{\pi(a|x)}{\sqrt{\pi_{\mathrm{ref}}(a|x)}} \sqrt{\pi_{\mathrm{ref}}(a|x)} f(x,a)$$

$$\leq \sqrt{\sum_{x \in \mathcal{X}} \nu_0(x) \sum_{a \in \mathcal{A}} \frac{\pi^2(a|x)}{\pi_{\mathrm{ref}}(a|x)}} \sqrt{\sum_{x \in \mathcal{X}} \nu_0(x) \sum_{a \in \mathcal{A}} \pi_{\mathrm{ref}}(a|x) f^2(x,a)}$$

$$\leq \sqrt{C^\pi} \sqrt{\mathrm{Err}_{\pi_{\mathrm{ref}}}(f)},$$

where the last inequality holds because, thanks to Jensen's inequality, we have that

$$\sum_{a \in \mathcal{A}} \pi_{\mathrm{ref}}(a|x)f^2(x,a) \leq \sum_{a \in \mathcal{A}} \sum_{b \in \mathcal{A}} \pi_{\mathrm{ref}}(b|x)\pi_{\mathrm{ref}}(a|x)(r^\star(x,a) - r^\star(x,b) - \bar{r}(x,a) + \bar{r}(x,b))^2.$$

Endowed with this result, we continue by upper bounding equation 20 as follows

$$\langle \pi^\star, r^\star - \bar{r} \rangle \leq \sqrt{C^\star \sum_{x \in \mathcal{X}} \nu_0(x) \sum_{a \in \mathcal{A}} \pi_{\mathrm{ref}}(a|x)\pi_{\mathrm{ref}}(b|x)(r^\star(x,a) - r^\star(x,b) - \bar{r}(x,a) + \bar{r}(x,b))^2}$$

$$+ \mathrm{Shift}$$

$$\leq \sqrt{C^\star \mathrm{Err}_{\pi_{\mathrm{ref}}}(\bar{r})} + \mathrm{Shift}$$

$$\leq \frac{\beta' C^\star}{2} + \frac{\mathrm{Err}_{\pi_{\mathrm{ref}}}(\bar{r})}{\beta'} + \mathrm{Shift}$$

where in the last line we recognized the generalization error $\mathrm{Err}_{\pi_{\mathrm{ref}}}(\bar{r})$ for the hypothesis $\bar{r}$ on the response distribution generated by $\pi_{\mathrm{ref}}$. In the above derivation, we added and subtracted the term $r^\star(s,b)$ because the Bradley-Terry preference distribution is invariant under a constant additional

shift of the reward function; therefore, we can only aim at learning the difference between the reward assigned to two distinct actions after seeing the same prompt. We can not learn the individual values $r^\star(s, a)$ and $r^\star(s, b)$.

Following the same steps, we can also bound the expected reward difference under $\pi_{\text{out}}$.

$$\langle \pi_{\text{out}}, \bar{r} - r^\star \rangle \le \sqrt{C^{\pi_{\text{out}}} \text{Err}_{\pi_{\text{ref}}}(\bar{r})} - \text{Shift}$$
$$\le \frac{\beta' C^{\pi_{\text{out}}}}{2} + \frac{\text{Err}_{\pi_{\text{ref}}}(\bar{r})}{\beta'} - \text{Shift}.$$

At this point, plugging into equation 19, we obtain

$$\langle \pi^\star, r^\star \rangle - \langle \pi_{\text{out}}, r^\star \rangle \le \langle \pi^\star, r - \bar{r} \rangle + \frac{\beta' \gamma}{2} C^\star + \beta' D_{\text{KL}}(\pi^\star, \pi_{\text{ref}}) + \langle \pi_{\text{out}}, \bar{r} - r \rangle$$

$$- \frac{\beta' \gamma}{2} C^{\pi_{\text{out}}} - \beta' D_{\text{KL}}(\pi_{\text{out}}, \pi_{\text{ref}})$$

$$\le \frac{\beta' \gamma C^\star}{2} + \frac{\text{Err}_{\pi_{\text{ref}}}(\bar{r})}{\beta'}$$

$$+ \frac{\beta' C^\star}{2} + \beta' D_{\text{KL}}(\pi^\star, \pi_{\text{ref}})$$

$$+ \frac{\beta' C^{\pi_{\text{out}}}}{2} + \frac{\text{Err}_{\pi_{\text{ref}}}(\bar{r})}{\beta'}$$

$$- \frac{\beta' \gamma}{2} C^{\pi_{\text{out}}} - \beta' D_{\text{KL}}(\pi_{\text{out}}, \pi_{\text{ref}})$$

At this point, let us the fact that $\frac{1}{2}(C^\pi - 1) = D_{\chi^2}(\pi, \pi_{\text{ref}}) \ge D_{\text{KL}}(\pi, \pi_{\text{ref}})$ for all policies $\pi$. Therefore, $D_{\text{KL}}(\pi^\star, \pi_{\text{ref}}) \le \frac{C^\star}{2}$. Using these two facts, we obtain

$$\langle \pi^\star, r^\star \rangle - \langle \pi_{\text{out}}, r^\star \rangle \le \frac{\beta' \gamma C^\star}{2} + \frac{\text{Err}_{\pi_{\text{ref}}}(\bar{r})}{\beta'}$$

$$+ \beta' C^\star$$

$$+ \beta' C^{\pi_{\text{out}}} \left( \frac{1}{2} - \frac{\gamma}{2} \right) + \frac{\text{Err}_{\pi_{\text{ref}}}(\bar{r})}{\beta'} - \beta' D_{\text{KL}}(\pi_{\text{out}}, \pi_{\text{ref}})$$

Therefore, if $\gamma = 1$, using that $\beta' D_{\text{KL}}(\pi_{\text{out}}, \pi_{\text{ref}}) \ge 0$, we have that

$$\langle \pi^\star, r^\star \rangle - \langle \pi_{\text{out}}, r^\star \rangle \le \frac{2\text{Err}_{\pi_{\text{ref}}}(\bar{r})}{\beta'} + 3\beta' C^\star$$

while for $\gamma = 0$,

$$\langle \pi^\star, r^\star \rangle - \langle \pi_{\text{out}}, r^\star \rangle \le \frac{2\text{Err}_{\pi_{\text{ref}}}(\bar{r})}{\beta'} + \frac{\beta'}{2}(C^\star + C^{\pi_{\text{out}}} - 2D_{\text{KL}}(\pi_{\text{out}}, \pi_{\text{ref}}))$$

$$\le \frac{2\text{Err}_{\pi_{\text{ref}}}(\bar{r})}{\beta'} + \max_{\pi \in \Pi'} \frac{\beta'}{2}(C^\star + C^\pi - 2D_{\text{KL}}(\pi, \pi_{\text{ref}}))$$

$$:= \frac{2\text{Err}_{\pi_{\text{ref}}}(\bar{r})}{\beta'} + \beta' \Delta D(\Pi')$$

**Case $\gamma = 1$.** Setting $\beta' = \sqrt{\frac{2\text{Err}_{\max}}{3C^\star}}$ for any $\text{Err}_{\max}$ such that $\text{Err}_{\pi_{\text{ref}}}(\bar{r}) \le \text{Err}_{\max}$ in high probability. Then, we have that

$$\langle \pi^\star, r^\star \rangle - \langle \pi_{\text{out}}, r^\star \rangle \le 8\sqrt{\text{Err}_{\max} C^\star}.$$

**Case $\gamma = 0$.** Setting $\beta' = \sqrt{\frac{2\text{Err}_{\max}}{\Delta D(\Pi')}}$, ensures that

$$\langle \pi^\star, r^\star \rangle - \langle \pi_{\text{out}}, r^\star \rangle \le 4\sqrt{\text{Err}_{\max} \Delta D(\Pi')}.$$

The final statement follows noticing that by definition, we have that $\hat{r} = \beta' \phi \left( \frac{\pi_{\text{out}}}{\pi_{\text{ref}}} \right)$. Then, by definition of $\text{Err}_{\pi_{\text{ref}}}(\bar{r})$, it holds that $\text{Err}_{\pi_{\text{ref}}}(\pi_{\text{out}}) = \text{Err}_{\pi_{\text{ref}}}(\bar{r})$. $\qquad \square$

The next step towards the proof of Theorem 4.1 is to decompose the generalization error of the mixed reward $\hat{r}$ into a contribution depending only on the ratings gap-based estimator and a second contribution depending only on the estimator learned via ranking.

**Lemma C.2.** *Error Decomposition Let the policy class $\Pi'$ and the errors $\mathrm{Err}_{\mathrm{DPO}}(N, \delta)$ and $\mathrm{Err}_{\pi_{\mathrm{ref}}}(\hat{r})$ be defined as in Theorem 4.1, then choosing $\alpha = (1 + \mathrm{Err}_{\pi_{\mathrm{ref}}}(\hat{r})/\mathrm{Err}_{\mathrm{DPO}}(N, \delta))^{-1}$ it holds that*

$$\mathrm{Err}_{\pi_{\mathrm{ref}}}(\bar{r}) \leq 2 \left( \mathrm{Err}_{\mathrm{DPO}}^{-1}(N, \delta) + \mathrm{Err}_{\pi_{\mathrm{ref}}}^{-1}(\hat{r}) \right)^{-1} \leq 2 \min \{ \mathrm{Err}_{\mathrm{DPO}}(N, \delta), \mathrm{Err}_{\pi_{\mathrm{ref}}}(\hat{r}) \}$$

*Proof.* Let us start replacing the definition of $\bar{r}$ in the definition of $\mathrm{Err}_{\pi_{\mathrm{ref}}}(\bar{r})$. We obtain,

$$\mathrm{Err}_{\pi_{\mathrm{ref}}}(\bar{r}) = \sum_{x \in \mathcal{X}} \nu_0(x) \sum_{a \in \mathcal{A}} \pi_{\mathrm{ref}}(a|x) \pi_{\mathrm{ref}}(b|x) \left( \Delta_{r^\star}(x, a, b) - \beta' \Delta_{\phi \pi_{\mathrm{out}}/\pi_{\mathrm{ref}}}(x, a, b) \right)^2$$

$$= \sum_{x \in \mathcal{X}} \nu_0(x) \sum_{a \in \mathcal{A}} \pi_{\mathrm{ref}}(a|x) \pi_{\mathrm{ref}}(b|x) \left( \Delta_{r^\star}(x, a, b) - \Delta_{(1-\alpha)r_{\mathrm{out}} + \alpha \hat{r}}(x, a, b) \right)^2$$

at this point, we use the following decompositions

$$\Delta_{(1-\alpha)r_{\mathrm{out}} + \alpha \hat{r}}(x, a, b) = (1 - \alpha)(r_{\mathrm{out}}(x, a) - r_{\mathrm{out}}(x, b)) + \alpha(\hat{r}(x, a) - \hat{r}(x, b))$$

$$= (1 - \alpha)\Delta_{r_{\mathrm{out}}}(x, a, b) + \alpha \Delta_{\hat{r}}(x, a, b)$$

Equipped of this fact and rewriting $\Delta_{r^\star}(x, a, b) = (1 - \alpha)\Delta_{r^\star}(x, a, b) + \alpha \Delta_{r^\star}(x, a, b)$, we have,

$$\mathrm{Err}_{\pi_{\mathrm{ref}}}(\bar{r}) = \sum_{x \in \mathcal{X}} \nu_0(x) \sum_{a \in \mathcal{A}} \pi_{\mathrm{ref}}(a|x) \pi_{\mathrm{ref}}(b|x) \Big( (1 - \alpha)(\Delta_{r^\star}(x, a, b) - \Delta_{r_{\mathrm{out}}}(x, a, b))$$

$$+ \alpha(\Delta_{r^\star}(x, a, b) - \Delta_{\hat{r}}(x, a, b)) \Big)^2$$

$$\leq 2(1 - \alpha)^2 \sum_{x \in \mathcal{X}} \nu_0(x) \sum_{a \in \mathcal{A}} \pi_{\mathrm{ref}}(a|x) \pi_{\mathrm{ref}}(b|x) (\Delta_{r^\star}(x, a, b) - \Delta_{r_{\mathrm{out}}}(x, a, b))^2$$

$$+ 2\alpha^2 \sum_{x \in \mathcal{X}} \nu_0(x) \sum_{a \in \mathcal{A}} \pi_{\mathrm{ref}}(a|x) \pi_{\mathrm{ref}}(b|x) (\Delta_{r^\star}(x, a, b) - \Delta_{\hat{r}}(x, a, b))^2$$

where the inequality follows from Young's inequality. The second term is by definition equal to $\mathrm{Err}_{\pi_{\mathrm{ref}}}(\hat{r})$. For the first term, we notice that we can rewrite the optimization problem in equation 2 as a log likelihood maximization problem over reward variables

$$r_{\mathrm{out}} = \underset{r \in \mathcal{R}_{\Pi'}}{\mathrm{argmax}} \sum_{i=1}^{N} \log \sigma \left( r(x_i, a_i^+) - r(x_i, a_i^-) \right)$$

for the reward class $\mathcal{R}_{\Pi'} = \left\{ r | \exists \pi \in \Pi' \text{ such that } r(x, a) = \frac{1}{1-\alpha} \left( \beta' \phi \left( \frac{\pi(a|x)}{\pi_{\mathrm{ref}}(a|x)} \right) - \alpha \hat{r}(x, a) \right) \right\}$. Here, the reward class depends on $\hat{r}$, which could be dependent on the dataset $\mathcal{D}$. This invalidates the classic MLE results (see e.g. (Foster & Rakhlin, 2023, Proposition 1)) if the dataset of rankings is correlated with the dataset of ratings. Notice that this is indeed the case if ratings and rankings come from the same source as for example in `ultrafeedback-binarized` dataset. However, we know that $\hat{r} \in \mathcal{R}$ by definition. Therefore, let us handle dependent data with a covering number. In particular, let us fix $\tilde{r} \in \mathcal{R}$ such that the policy induced reward class $\mathcal{R}_{\tilde{r}}$ defined as

$$\mathcal{R}_{\tilde{r}} = \left\{ r | \exists \pi \in \Pi'_{\tilde{r}} \text{ such that } r(x, a) = \frac{1}{1 - \alpha} \left( \beta' \phi \left( \frac{\pi(a|x)}{\pi_{\mathrm{ref}}(a|x)} \right) - \alpha \tilde{r}(x, a) \right) \right\}. \quad (21)$$

with $\Pi'_{\tilde{r}}$ being the analog of $\Pi'$ where the fixed reward $\tilde{r}$ replaces $\hat{r}$. That is,

$$\Pi'_{\tilde{r}} = \left\{ \pi \in \Pi : \text{s.t. } \forall x, a, b \ \left| \beta' \Delta_{\phi \pi/\pi_{\mathrm{ref}}}(x, a, b) - \alpha \Delta_{\tilde{r}}(x, a, b) \right| \leq (1 - \alpha)(R_{\max} - R_{\min}) \right\},$$

where $\Delta_{\tilde{r}}(x, a, b) = \tilde{r}(x, a) - \tilde{r}(x, b)$. Clearly, $\mathcal{R}_{\tilde{r}}$ is independent of $\mathcal{D}$ since $\tilde{r}$ is independent of $\mathcal{D}$ [7]. Next, we show that this reward class realizes the true reward $r^\star$ up to a state-dependent shift $\zeta : \mathcal{X} \to \mathbb{R}$. To this end let us define the policy $\pi_{\tilde{r}}$ as follows

$$\pi_{\tilde{r}}(a|x) = \pi_{\text{ref}}(a|x)\phi^{-1}(\beta'^{-1}((1-\alpha)r^\star(x, a) + \alpha\tilde{r}(x, a) + \zeta(x)))$$

where $\zeta(x)$ is chosen to ensure that $\sum_{a \in \mathcal{A}} \pi_{\tilde{r}}(a|x) = 1$ for all $x \in \mathcal{X}$. Now we show that $\pi_{\tilde{r}} \in \Pi'_{\tilde{r}}$. Rearranging, we have that,

$$\beta'\phi\left(\frac{\pi_{\tilde{r}}(a|x)}{\pi_{\text{ref}}(a|x)}\right) = (1-\alpha)r^\star(x, a) + \alpha\tilde{r}(x, a) + \zeta(x)$$

Therefore, taking the difference between the log policy ratios evaluated at two different actions $a, b$ for a fixed prompt $x$, we have that

$$\beta'\phi\left(\frac{\pi_{\tilde{r}}(a|x)}{\pi_{\text{ref}}(a|x)}\right) - \beta'\phi\left(\frac{\pi_{\tilde{r}}(b|x)}{\pi_{\text{ref}}(b|x)}\right) = (1-\alpha)(r^\star(x, a) - r^\star(x, b)) + \alpha(\tilde{r}(x, a) - \tilde{r}(x, b)).$$

It follows that

$$(1-\alpha)(R_{\min} - R_{\max}) + \alpha(\tilde{r}(x, a) - \tilde{r}(x, b)) \le \beta'\phi\left(\frac{\pi_{\tilde{r}}(a|x)}{\pi_{\text{ref}}(a|x)}\right) - \beta'\phi\left(\frac{\pi_{\tilde{r}}(b|x)}{\pi_{\text{ref}}(b|x)}\right)$$
$$\le (1-\alpha)(R_{\max} - R_{\min}) + \alpha(\tilde{r}(x, a) - \tilde{r}(x, b)).$$

Therefore, for any $\tilde{r} \in \mathcal{R}$, we have that $\pi_{\tilde{r}} \in \Pi'_{\tilde{r}}$. At this point, notice that for $\pi(a|x) = \pi_{\tilde{r}}(a|x)$ in equation 21, we have that the true reward plus a state (prompt) dependent shift belongs to the induced reward class. In formulas, we have that there exists $\zeta : \mathcal{X} \to \mathbb{R}$ such that $r_\zeta \in \mathcal{R}_{\tilde{r}}$ where we defined $r_\zeta(x, a) = r(x, a) + \zeta(x)/(1-\alpha)$ for all $x, a$. We can conclude that the true reward function is realizable (up to a state dependent shift) for all possible values of $\tilde{r}$. Introducing the probability model, $P_r(a \text{ is preferred to } b|x, a, b) = \sigma(r(x, a) - r(x, b))$, we can consider that

$$r_{\text{out}} = \underset{r \in \mathcal{R}_{\tilde{r}}}{\operatorname{argmax}} \sum_{i=1}^{N_{\text{rank}}} \log P_r(a_i^+ \text{ is preferred to } a_i^- | x_i, a_i^-, a_i^+).$$

Notice that since $r_\zeta \in \mathcal{R}_{\tilde{r}}$ we have that

$$P_{r_\zeta}(a \text{ is preferred to } b|x, a, b) = \sigma\left(r^\star(x, a) + \zeta(x)/(1-\alpha) - r^\star(x, b) - \zeta(x)/(1-\alpha)\right)$$
$$= \sigma\left(r^\star(x, a) - r^\star(x, b)\right)$$
$$= P_{r^\star}(a \text{ is preferred to } b|x, a, b)$$

Therefore, the true preference generator $P_{r^\star}$ is realized by the preference models class induced by the reward class $\mathcal{R}_{\tilde{r}}$ even if we do not necessarily have that $r^\star \in \mathcal{R}_{\tilde{r}}$. This fact should not surprise the reader since the Bradley-Terry model is invariant to shifts depending on the state only.

Moreover, the definition of $\Pi'$ guarantees that

$$\pi \in \Pi' \implies \left|\beta'\Delta_{\phi\pi/\pi_{\text{ref}}}(x, a, b) - \alpha\Delta_{\tilde{r}}(x, a, b)\right| \le (1-\alpha)(R_{\max} - R_{\min}) \quad \forall x, a, b.$$

Therefore, since $\pi_{\text{out}} \in \Pi'$, the above fact implies that $\Delta_{r_{\text{out}}} \in [-R_{\max}, R_{\max}]$. At this point, thanks to Lemma E.2 , a covering number over the class $\mathcal{R}$ and with Lemma E.3 invoked with failure probability $\delta/|\mathcal{R}|$, we obtain that with probability at least $1 - \delta$ it holds that

$$(1-\alpha)^2 \sum_{x \in \mathcal{X}} \nu_0(x) \sum_{a \in \mathcal{A}} \pi_{\text{ref}}(a|x)\pi_{\text{ref}}(b|x) \left(\Delta_{r^\star}(x, a, b) - \Delta_{r_{\text{out}}}(x, a, b)\right)^2$$
$$= (1-\alpha)^2 \sum_{x \in \mathcal{X}} \nu_0(x) \sum_{a \in \mathcal{A}} \pi_{\text{ref}}(a|x)\pi_{\text{ref}}(b|x) \left(r^\star(x, a) - r^\star(x, b) - r_{\text{out}}(a|x) + r_{\text{out}}(x, b)\right)^2$$
$$\le 32R_{\max}^2(1-\alpha)^2 e^{4R_{\max}} \frac{\log(|\Pi'| \, |\mathcal{R}| \, /\delta)}{N_{\text{rank}}}$$
$$= \text{Err}_{\text{DPO}}(N, \delta),$$

---

[7] in fact, it is a fixed quantity, not a random variable.

where the last equality holds by definition of $\text{Err}_{\text{DPO}}(N, \delta)$ and by the fact that $\Pi' \subset \Pi$. Putting all together, we have that

$$\text{Err}_{\pi_{\text{ref}}}(\bar{r}) \leq 2\left((1-\alpha)^2 \text{Err}_{\text{DPO}}(N,\delta) + \alpha^2 \text{Err}(\hat{r})\right)$$

$$= \frac{2\text{Err}_{\text{DPO}}(N,\delta)\text{Err}(\hat{r})}{\text{Err}_{\text{DPO}}(N,\delta) + \text{Err}(\hat{r})}$$

where the first equality follows from the choice of $\alpha$. Finally, using the relation $(1/\mathfrak{a} + 1/\mathfrak{b})^{-1} \leq \min(\mathfrak{a}, \mathfrak{b})$ that holds between any two positive scalars $\mathfrak{a}, \mathfrak{b} \in \mathbb{R}^+$, we that

$$\left(\text{Err}_{\text{DPO}}^{-1}(N,\delta) + \text{Err}_{\pi_{\text{ref}}}^{-1}(\hat{r})\right)^{-1} \leq \min\left\{\text{Err}_{\text{DPO}}(N,\delta), \text{Err}_{\pi_{\text{ref}}}(\hat{r})\right\} \tag{22}$$

and therefore

$$\text{Err}_{\pi_{\text{ref}}}(\bar{r}) \leq 2\min\left\{\text{Err}_{\text{DPO}}(N,\delta), \text{Err}_{\pi_{\text{ref}}}(\hat{r})\right\}.$$

$\square$

Now we can state the proof of Theorem 4.1.

*Proof. of Theorem 4.1* Using Lemma C.2, we can set $\text{Err}_{\max}$ in Lemma C.1 as

$$\text{Err}_{\max} = \frac{2\text{Err}_{\text{DPO}}(N,\delta)\text{Err}(\hat{r})}{\text{Err}_{\text{DPO}}(N,\delta) + \text{Err}(\hat{r})}.$$

This concludes the proof. $\square$

## C.3 PROOF OF THEOREM 4.2

We prove the following more general version, which allows us to consider an additional $\chi^2$ regularization term.

**Theorem C.2.** *Let us assume that we have conditional independence between $z_i$ and $\Delta_{\hat{r}}^i$ given $x_i, a_i, a_i'$, and rating gaps are Gaussian $\Delta_{\hat{r}}^i \sim \mathcal{N}(\Delta_r^i, \mathbb{V})$, for all $i \in [N]$. Under these condition, let $\pi_{\text{out}}$ be a solution to Eq. 16 optimized over a policy class $\tilde{\Pi} \subset \Pi$ such that $\pi \in \tilde{\Pi}$ implies $\beta\Delta_{\log \pi/\pi_{\text{ref}}} \in [-R_{\max}, R_{\max}]$ with $\beta = \tilde{\mathcal{O}}\left(\sqrt{\frac{\min\{R_{\max}^2 e^{R_{\max}}, \mathbb{V}+R_{\max}^2\}\log(|\Pi|/\delta)}{C^\star N}}\right)$ and $\phi(x) = \log(x)$. Then, with probability $1 - \delta$, the sub-optimality of $\pi_{\text{out}}$ i.e., $\langle \pi^\star - \pi_{\text{out}}, r^\star \rangle$, is upper bounded by $\tilde{\mathcal{O}}\left(\sqrt{\frac{\Delta D(\tilde{\Pi})\min\{e^{R_{\max}}R_{\max}^2, R_{\max}^2+\mathbb{V}\}\log(|\Pi|/\delta)}{N}}\right)$. Moreover, if we consider $\pi_{\text{out}}$ as the minimizer of Equation (16) using $\phi(x) = \phi_{\text{KL}+\chi^2}(x) = x + \log(x)$ we obtain that with probability $1 - \delta$,*

$$\langle \pi^\star - \pi_{\text{out}}, r^\star \rangle \leq \tilde{\mathcal{O}}\left(\sqrt{\frac{C^\star \min\{e^{R_{\max}}R_{\max}^2, R_{\max}^2+\mathbb{V}\}\log(|\Pi|/\delta)}{N}}\right).$$

*Proof.* Let us introduce $r_{\text{out}} = \beta\phi\left(\frac{\pi_{\text{out}}}{\pi_{\text{ref}}}\right)$ where $\pi_{\text{out}}$ is the output policy of the algorithm computed via Eq. ML-RDPO. Therefore, via Lemma C.1 invoked for $\phi(x) = \phi_{\text{KL}+\chi^2}(x)$, it holds

$$\langle \pi^\star, r^\star \rangle - \langle \pi_{\text{out}}, r^\star \rangle \leq 8\sqrt{\text{Err}_{\pi_{\text{ref}}}(\pi_{\text{out}})C^\star}. \tag{23}$$

Moreover, invoking Lemma C.1 for $\phi(x) = \log(x)$, we get

$$\langle \pi^\star, r^\star \rangle - \langle \pi_{\text{out}}, r^\star \rangle \leq 8\sqrt{\text{Err}_{\pi_{\text{ref}}}(\pi_{\text{out}})\Delta D(\tilde{\Pi})}. \tag{24}$$

At this point, the proofs of RDPO and ML-RDPO now differ only in the technique used for bounding $\text{Err}_{\pi_{\text{ref}}}(\pi_{\text{out}})$. To this end, let us introduce the reward class $\mathcal{R}_{\tilde{\Pi}}$ as follows

$$\mathcal{R}_{\tilde{\Pi}} = \left\{r \mid r = \beta\phi\left(\frac{\pi_{\text{out}}}{\pi_{\text{ref}}}\right) \quad \text{s.t.} \quad \pi \in \tilde{\Pi}\right\},$$

where we recall that $\tilde{\Pi}$ is introduced in Section 3.2. It is important to notice that, by definition, of $\tilde{\Pi}$, we have that for any $r \in \mathcal{R}_{\tilde{\Pi}}$, it holds $\Delta_r \in [-R_{\max}, R_{\max}]$.

Endowed with these definitions, we can interpret finding the output of Eq. ML-RDPO as implicitly computing the following MLE

$$r_{\text{out}} = \operatorname*{argmax}_{r \in \mathcal{R}_{\tilde{\Pi}}} \log P_r^{\text{rat,rank}}(\mathcal{D}_{\text{rat}}, \mathcal{D}_{\text{rank}})$$

By the general bound for the conditional MLE risk Lemma E.3, we have that for the above MLE problem, we have that it holds that with probability $1 - \delta$

$$\sum_{x \in \mathcal{X}} \nu_0(x) \sum_{a,b \in \mathcal{A} \times \mathcal{A}} \pi_{\text{ref}}(a|x)\pi_{\text{ref}}(b|x) \int_{z \in \mathcal{Z}} \left( \sqrt{P_{r_{\text{out}}}^{\text{rat,rank}}(z|x,a,b)} - \sqrt{P_{r^\star}^{\text{rat,rank}}(z|x,a,b)} \right)^2$$

$$\leq \frac{2 \log(|\Pi'|/\delta)}{N}.$$

where $N$ is the total number of feedback received ( ratings and feedback) and $\mathcal{Z}$ is the set of all possible feedbacks, that is $\mathcal{Z} = \{0,1\} \times \mathbb{R}$. Under the conditional independence assumption, we have that by the data processing inequality for any $\alpha \in [0,1]$

$$(1 - \alpha) \sum_{x \in \mathcal{X}} \nu_0(x) \sum_{a,b \in \mathcal{A} \times \mathcal{A}} \pi_{\text{ref}}(a|x)\pi_{\text{ref}}(b|x) \sum_{z \in \{0,1\}} \left( \sqrt{P_{r_{\text{out}}}^{\text{rank}}(z|x,a,b)} - \sqrt{P_{r^\star}^{\text{rank}}(z|x,a,b)} \right)^2$$

$$+ \alpha \sum_{x \in \mathcal{X}} \nu_0(x) \sum_{a,b \in \mathcal{A} \times \mathcal{A}} \pi_{\text{ref}}(a|x)\pi_{\text{ref}}(b|x) \int_{-\infty}^{\infty} \left( \sqrt{P_{r_{\text{out}}}^{\text{rat}}(z|x,a,b)} - \sqrt{P_{r^\star}^{\text{rat}}(z|x,a,b)} \right)^2 dz$$

$$\leq \frac{2 \log(|\Pi'|/\delta)}{N}.$$

By Lemma E.2, it holds that

$$\sum_{z \in \{0,1\}} \left( \sqrt{P_{r_{\text{out}}}^{\text{rank}}(z|x,a,b)} - \sqrt{P_{r^\star}^{\text{rank}}(z|x,a,b)} \right)^2 \geq \frac{(r_{\text{out}}(x,a) - r_{\text{out}}(x,b) - r^\star(x,a) - r^\star(x,b))^2}{16e^{4R_{\max}}R_{\max}^2}.$$

Finally, since $P_r^{\text{rat}}$ is Gaussian for any $r \in \mathcal{R}$ with same variance $\mathbb{V}$, we have that

$$\int_{-\infty}^{\infty} \left( \sqrt{P_{r_{\text{out}}}^{\text{rat}}(z|x,a,b)} - \sqrt{P_{r^\star}^{\text{rat}}(z|x,a,b)} \right)^2 dz$$

$$= 2 - 2 \exp \left( \frac{-(r_{\text{out}}(x,a) - r_{\text{out}}(x,b) - r^\star(x,a) - r^\star(x,b))^2}{8\mathbb{V}} \right)$$

$$\geq \frac{(r_{\text{out}}(x,a) - r_{\text{out}}(x,b) - r^\star(x,a) - r^\star(x,b))^2}{4\mathbb{V}((r_{\text{out}}(x,a) - r_{\text{out}}(x,b) - r^\star(x,a) - r^\star(x,b))^2/(8\mathbb{V}) + 1)}$$

$$\geq \frac{(r_{\text{out}}(x,a) - r_{\text{out}}(x,b) - r^\star(x,a) - r^\star(x,b))^2}{4\mathbb{V}((R_{\max} - R_{\min})^2/(2\mathbb{V}) + 1)}$$

$$\geq \frac{(r_{\text{out}}(x,a) - r_{\text{out}}(x,b) - r^\star(x,a) - r^\star(x,b))^2}{2(R_{\max} - R_{\min})^2 + 4\mathbb{V}},$$

where the first equality holds because of the closed form solution for the squared Hellinger distance between Gaussian distributions with the same variance and different means [8] and the first inequality holds because $1 - \exp(-x) \geq x/x+1$ for all $x \geq 0$. All in all, recalling that we denote $\sum_{x \in \mathcal{X}} \nu_0(x) \sum_{a,b \in \mathcal{A} \times \mathcal{A}} \pi_{\text{ref}}(a|x)\pi_{\text{ref}}(b|x)(r_{\text{out}}(x,a) - r_{\text{out}}(x,b) - r^\star(x,a) - r^\star(x,b))^2$ by $\text{Error}_{\pi_{\text{ref}}}(\pi_{\text{out}})$ we get that

$$\left( \frac{1 - \alpha}{16e^{4R_{\max}}R_{\max}^2} + \frac{\alpha}{2(R_{\max} - R_{\min})^2 + 4\mathbb{V}} \right) \text{Error}_{\pi_{\text{ref}}}(\pi_{\text{out}}) \leq \frac{2 \log(\Pi'/\delta)}{N}$$

---
[8] https://en.wikipedia.org/wiki/Hellinger_distance

Therefore, we conclude that with probability $1 - \delta$,

$$\text{Error}_{\pi_{\text{ref}}}(\pi_{\text{out}}) \leq \left( \frac{16e^{4R_{\max}} R_{\max}^2 (2(R_{\max} - R_{\min})^2 + 4\mathbb{V})}{(2(R_{\max} - R_{\min})^2 + 4\mathbb{V})(1 - \alpha) + 16e^{4R_{\max}} R_{\max}^2 \alpha} \right) \frac{2 \log(\Pi'/\delta)}{N}.$$

At this point, choosing $\alpha = \mathbb{1}\left\{ 2(R_{\max} - R_{\min})^2 + 4\mathbb{V} \leq 16e^{4R_{\max}} R_{\max}^2 \right\}$, we have that

$$\text{Error}_{\pi_{\text{ref}}}(\pi_{\text{out}}) \leq \left( \frac{16e^{4R_{\max}} R_{\max}^2 (2(R_{\max} - R_{\min})^2 + 4\mathbb{V})}{\max\left\{ 2(R_{\max} - R_{\min})^2 + 4\mathbb{V}, 16e^{4R_{\max}} R_{\max}^2 \right\}} \right) \frac{2 \log(\Pi'/\delta)}{N}$$

$$= \min\left\{ 2(R_{\max} - R_{\min})^2 + 4\mathbb{V}, 16e^{4R_{\max}} R_{\max}^2 \right\} \frac{2 \log(\Pi'/\delta)}{N}.$$

Plugging back into Eq. 23 concludes the proof. □

# D   CONVERGENCE GUARANTEES FOR THE HETEROGENEOUS CASE

In this section, we report the convergence guarantees for the case where we observe either completely disjoint or only partly overlapping rating and ranking datasets. The variants of ML-RDPO and RDPO that apply to this setting are derived in Appendix B.2.

## D.1   CONVERGENCE GUARANTEES FOR RDPO IN THE HETEROGENEOUS CASE (ALGORITHM 2)

We start by stating the convergence guarantees for RDPO in the heterogeneous setting.

**Theorem D.1.** *Let us run RDPO for disjoint rating and ranking datasets ( i.e. Algorithm 2) with* $\gamma = 1$, $\alpha = \frac{R_{\max} e^{2R_{\max}} N_{\text{rat}}^{1/2}}{(R_{\max} + \sqrt{\mathbb{V} \log(2N_{\text{rat}}/\delta)}) N_{\text{rank}}^{1/2} + R_{\max} e^{2R_{\max}} N_{\text{rat}}^{1/2}}$ *and*

$$\beta = \sqrt{\frac{2}{3C^\star} \frac{R_{\max}^2 e^{4R_{\max}} (R_{\max} + \sqrt{\mathbb{V} \log(2N_{\text{rat}}/\delta)})^2 \log(|\mathcal{R}| (|\mathcal{R}| + |\Pi'|)/\delta)}{R_{\max}^2 e^{4R_{\max}} N_{\text{rat}} + (R_{\max} + \sqrt{\mathbb{V} \log(2N_{\text{rat}}/\delta)})^2 N_{\text{rank}}}}.$$

*Moreover, let us recall that* $\Pi' \subset \Pi$ *the policy class subset such that* $\pi \in \Pi' \implies \beta \Delta_{\phi \pi / \pi'_{\text{ref}}} \in [-R_{\max}, R_{\max}]$. *Then, it holds that* $\langle \pi^\star, r^\star \rangle - \langle \pi_{\text{out}}, r^\star \rangle$ *is upper bounded by*

$$\mathcal{O}\left( \sqrt{\frac{R_{\max}^2 e^{4R_{\max}} (R_{\max} + \sqrt{\mathbb{V} \log(2N_{\text{rat}}/\delta)})^2 \log(|\mathcal{R}| (|\mathcal{R}| + |\Pi'|)/\delta)}{R_{\max}^2 e^{4R_{\max}} N_{\text{rat}} + (R_{\max} + \sqrt{\mathbb{V} \log(2N_{\text{rat}}/\delta)})^2 N_{\text{rank}}} C^\star} \right),$$

*with probability at least* $1 - 3\delta$. *Moreover, running with* $\gamma = 0$ *and*

$$\beta = \sqrt{\frac{\sqrt{2}}{3\Delta D(\pi^\star, \pi_{\text{ref}})} \frac{R_{\max}^2 e^{4R_{\max}} (R_{\max} + \sqrt{\mathbb{V} \log(2N_{\text{rat}}/\delta)})^2 \log(|\mathcal{R}| (|\mathcal{R}| + |\Pi'|)/\delta)}{R_{\max}^2 e^{4R_{\max}} N_{\text{rat}} + (R_{\max} + \sqrt{\mathbb{V} \log(2N_{\text{rat}}/\delta)})^2 N_{\text{rank}}}},$$

*it holds that* $\langle \pi^\star, r^\star \rangle - \langle \pi_{\text{out}}, r^\star \rangle$ *is upper bounded by*

$$\mathcal{O}\left( \sqrt{\frac{R_{\max}^2 e^{4R_{\max}} (R_{\max} + \sqrt{\mathbb{V} \log(2N_{\text{rat}}/\delta)})^2 \log(|\mathcal{R}| (|\mathcal{R}| + |\Pi'|)/\delta)}{R_{\max}^2 e^{4R_{\max}} N_{\text{rat}} + (R_{\max} + \sqrt{\mathbb{V} \log(2N_{\text{rat}}/\delta)})^2 N_{\text{rank}}} \Delta D(\pi^\star, \pi_{\text{ref}})} \right),$$

*with probability at least* $1 - 3\delta$.

In the next subsection, we provide the proofs of both statements in a unified manner.

*Proof.  of Theorem D.1* First notice that in the heterogeneous case $\Delta_{\hat{r}}^i$ is replaced by $\Delta_{\hat{r}_{\text{rat}}}^i$. Therefore, using Theorem C.1 for $\phi(x) = \phi_{\text{KL}+\chi^2}(x)$ ( i.e. $\gamma = 1$) with $\text{Err}_{\hat{r}_{\text{rat}}}$ replacing $\text{Err}_{\hat{r}}$, we have that with probability at least $1 - \delta$,

$$\langle \pi^\star, r^\star \rangle - \langle \pi_{\text{out}}, r^\star \rangle \leq \mathcal{O}\left( \sqrt{C^\star \left( \text{Err}_{\text{DPO}}^{-1}(N, \delta) + \text{Err}_{\pi_{\text{ref}}}^{-1}(\hat{r}_{\text{rat}}) \right)^{-1}} \right)$$

$$= \mathcal{O}\left( \sqrt{C^\star \frac{\text{Err}_{\text{DPO}}(N, \delta) \text{Err}_{\pi_{\text{ref}}}(\hat{r}_{\text{rat}})}{\text{Err}_{\text{DPO}}(N, \delta) + \text{Err}_{\pi_{\text{ref}}}(\hat{r}_{\text{rat}})}} \right).$$

Then replacing, the expression for $\mathrm{Err}_{\mathrm{DPO}}(N, \delta)$, we obtain

$$\langle \pi^\star, r^\star \rangle - \langle \pi_{\mathrm{out}}, r^\star \rangle \leq \mathcal{O} \left( \sqrt{C^\star \frac{\frac{32 R_{\max}^2 e^{4R_{\max}} \log(|\Pi||\mathcal{R}|/\delta)}{N} \mathrm{Err}_{\pi_{\mathrm{ref}}}(\hat{r}_{\mathrm{rat}})}{\frac{32 R_{\max}^2 e^{4R_{\max}} \log(|\Pi||\mathcal{R}|/\delta)}{N} + \mathrm{Err}_{\pi_{\mathrm{ref}}}(\hat{r}_{\mathrm{rat}})}} \right).$$

Next, in order to bound $\mathrm{Err}_{\pi_{\mathrm{ref}}}(\hat{r}_{\mathrm{rat}})$, we invoke a standard result concerning the generalization error of the empirical risk minimization for the square loss with subgaussian noise ( see Lemma E.4) to obtain that with probability at least $1 - 2\delta$

$$\mathrm{Err}_{\pi_{\mathrm{ref}}}(\hat{r}_{\mathrm{rat}}) = \sum_{x \in \mathcal{X}} \nu_0(x) \sum_{a \in \mathcal{A}} \pi_{\mathrm{ref}}(a|x) \pi_{\mathrm{ref}}(b|x) \left( r^\star(x, a) - r^\star(x, b) - \hat{r}_{\mathrm{rat}}(x, a) + \hat{r}_{\mathrm{rat}}(x, b) \right)^2$$

$$\leq \frac{c(R_{\max} - R_{\min} + \sqrt{\mathbb{V} \log(2N_{\mathrm{rat}}/\delta)})^2 \log(|\mathcal{R}|/\delta)}{N_{\mathrm{rat}}}$$

$$\leq \frac{c(R_{\max} + \sqrt{\mathbb{V} \log(2N_{\mathrm{rat}}/\delta)})^2 \log(|\mathcal{R}|/\delta)}{N_{\mathrm{rat}}},$$

for some $c \in \mathbb{R}$. Now, plugging in this bound and rearranging yields the conclusions for the statement with $\phi(x) = \phi_{\mathrm{KL}+\chi^2}(x)$. The proof for $\phi(x) = \log(x)$ is analogous invoking Theorem C.1 for $\gamma = 0$. $\qquad \square$

## D.2 Convergence guarantees for ML-RDPO in the heterogeneous case

For the guarantees of ML-RDPO, let us consider that the datasets $\mathcal{D}_{\mathrm{rat}}$ and $\mathcal{D}_{\mathrm{rank}}$ are generated as follows. First, $N$ state action pairs are collected offline from $\pi_{\mathrm{ref}}$. For each of these pairs, we observe a rating with probability $p_{\mathrm{rat,obs}}$ and a ranking with probability $p_{\mathrm{rank,obs}}$. If a ranking is observed, the state action pair is added to $\mathcal{D}_{\mathrm{rank}}$. Similarly, if a rating is observed, the state action pair is appended to $\mathcal{D}_{\mathrm{rat}}$. Therefore, state action pairs might appear in both datasets or only in one of the two.

**Theorem D.2.** *Let us consider $\pi_{\mathrm{out}}$ computed as in Eq. 17 with $\gamma = 1$ and*

$$\beta = \sqrt{\frac{2}{3C^\star}} \sqrt{\min \left\{ \frac{(2(R_{\max} - R_{\min})^2 + 4\mathbb{V})}{p_{\mathrm{obs,rat}}}, \frac{16 e^{4R_{\max}} R_{\max}^2}{p_{\mathrm{obs,rank}}} \right\} \frac{2 \log(|\Pi|/\delta)}{N}}.$$

*Then, it holds that with probability $1 - \delta$,*

$$\langle \pi^\star, r^\star \rangle - \langle \pi_{\mathrm{out}}, r^\star \rangle \leq \mathcal{O} \left( \sqrt{C^\star \min \left\{ \frac{(2(R_{\max} - R_{\min})^2 + 4\mathbb{V})}{p_{\mathrm{obs,rat}}}, \frac{16 e^{4R_{\max}} R_{\max}^2}{p_{\mathrm{obs,rank}}} \right\} \frac{2 \log(|\Pi|/\delta)}{N}} \right)$$

$$= \mathcal{O} \left( \sqrt{C^\star \min \left\{ \frac{(2(R_{\max} - R_{\min})^2 + 4\mathbb{V})}{\mathbb{E}[N_{\mathrm{rat}}]}, \frac{16 e^{4R_{\max}} R_{\max}^2}{\mathbb{E}[N_{\mathrm{rank}}]} \right\} 2 \log(|\Pi|/\delta)} \right),$$

*where we recall that $C^\star = \sum_{x \in \mathcal{X}} \nu_0(x) \sum_{a \in \mathcal{A}} \pi^\star(a|x) \left| \frac{\pi^\star(a|x)}{\pi_{\mathrm{ref}}(a|x)} \right|$.*

*Proof.* **Proof of Theorem D.2** For any $r \in \mathcal{R}_{\bar{\Pi}}$, let us define the ratings and ranking probability law to take into account the possibility of not observing a rating or a ranking, respectively. In particular, let us denote by $\emptyset$ the event where the rating or the ranking is not observed. Then, we define

$$\tilde{P}_r^{\mathrm{rat}}(z|x, a, b) = \begin{cases} P_r^{\mathrm{rat}}(z|x, a, b) p_{\mathrm{obs,rat}} & \text{if } z \neq \emptyset \\ 1 - p_{\mathrm{obs,rat}} & \text{otherwise} \end{cases}$$

Let us define the policy class and

$$\tilde{P}_r^{\mathrm{rank}}(z|x, a, b) = \begin{cases} P_r^{\mathrm{rank}}(z|x, a, b) p_{\mathrm{obs,rank}} & \text{if } z \neq \emptyset \\ 1 - p_{\mathrm{obs,rank}} & \text{otherwise} \end{cases}$$

ML-RDPO can therefore be seen as the algorithm seeking implicitly the maximizer of the following loglikelihood problem

$$r_{\mathrm{out}} = \operatorname*{argmax}_{r \in \mathcal{R}_{\bar{\Pi}}} \log \tilde{P}_r^{\mathrm{rat,rank}}(\mathcal{D}_{\mathrm{rat}}, \mathcal{D}_{\mathrm{rank}})$$

$$= \operatorname*{argmax}_{r \in \mathcal{R}_{\bar{\Pi}}} \sum_{i=1}^{N} \log(\tilde{P}_r^{\mathrm{rat}}(\Delta_{\hat{r}}^i | x_i', \tilde{a}_i, \bar{a}_i) \tilde{P}_r^{\mathrm{rank}}(z_i | x_i, a_i, a_i'))$$

where $\Delta_r^i$'s and $z_i$'s are possibly equal to $\emptyset$. Therefore, let us consider the squared Hellinger divergence

$$D_H^2(p, q) = \int_{z \in \mathcal{Z}} \left( \sqrt{p(z|x, a, b)} - \sqrt{q(z|x, a, b)} \right)^2$$

via the general bound for the conditional MLE risk, we have that with probability $1 - \delta$,

$$\sum_{x \in \mathcal{X}} \nu_0(x) \sum_{a,b \in \mathcal{A} \times \mathcal{A}} \pi_{\text{ref}}(a|x)\pi_{\text{ref}}(b|x) \int_{z \in \mathcal{Z}} \left( \sqrt{\tilde{P}_{r_{\text{out}}}^{\text{rat,rank}}(z|x, a, b)} - \sqrt{\tilde{P}_{r^\star}^{\text{rat,rank}}(z|x, a, b)} \right)^2$$

$$\leq \frac{2\log(|\Pi|/\delta)}{N},$$

where we also upper bounded the size of the hypothesis class using the fact that

$$\left| \mathcal{R}_{\tilde{\Pi}} \right| \leq \left| \tilde{\Pi} \right| \leq |\Pi|.$$

At this point, due to the data processing inequality for the squared Hellinger divergence, we have that for all $x, a, b \in \mathcal{X} \times \mathcal{A} \times \mathcal{A}$ it holds that

$$D_H^2(\tilde{P}_{r_{\text{out}}}^{\text{rat,rank}}(\cdot|x, a, b), \tilde{P}_{r^\star}^{\text{rat,rank}}(\cdot|x, a, b)) \geq D_H^2(\tilde{P}_{r_{\text{out}}}^{\text{rat}}(\cdot|x, a, b), \tilde{P}_{r^\star}^{\text{rat}}(\cdot|x, a, b))$$

and

$$D_H^2(\tilde{P}_{r_{\text{out}}}^{\text{rat,rank}}(\cdot|x, a, b), \tilde{P}_{r^\star}^{\text{rat,rank}}(\cdot|x, a, b)) \geq D_H^2(\tilde{P}_{r_{\text{out}}}^{\text{rank}}(\cdot|x, a, b), \tilde{P}_{r^\star}^{\text{rank}}(\cdot|x, a, b)).$$

Therefore, for any $\alpha \in [0, 1]$ we have that

$$(1-\alpha)D_H^2(\tilde{P}_{r_{\text{out}}}^{\text{rank}}(\cdot|x, a, b), \tilde{P}_{r^\star}^{\text{rank}}(\cdot|x, a, b)) + \alpha D_H^2(\tilde{P}_{r_{\text{out}}}^{\text{rat}}(\cdot|x, a, b), \tilde{P}_{r^\star}^{\text{rat}}(\cdot|x, a, b)) \leq \frac{2\log(|\Pi|/\delta)}{N}$$

At this point, let us connect the squared Hellinger divergence $D_H^2(\tilde{P}_{r_{\text{out}}}^{\text{rat}}(\cdot|x, a, b), \tilde{P}_{r^\star}^{\text{rat}}(\cdot|x, a, b))$ with $D_H^2(P_{r_{\text{out}}}^{\text{rat}}(\cdot|x, a, b), P_{r^\star}^{\text{rat}}(\cdot|x, a, b))$.

$$D_H^2(\tilde{P}_{r_{\text{out}}}^{\text{rat}}(\cdot|x, a, b), \tilde{P}_{r^\star}^{\text{rat}}(\cdot|x, a, b)) = \int_{z \in \mathcal{Z}\backslash\{\emptyset\}} \left( \sqrt{\tilde{P}_{r_{\text{out}}}^{\text{rat}}(z|x, a, b)} - \sqrt{\tilde{P}_{r^\star}^{\text{rat}}(z|x, a, b)} \right)^2$$

$$+ \left( \sqrt{\tilde{P}_{r_{\text{out}}}^{\text{rat}}(\emptyset|x, a, b)} - \sqrt{\tilde{P}_{r^\star}^{\text{rat}}(\emptyset|x, a, b)} \right)^2$$

$$= \int_{z \in \mathcal{Z}\backslash\{\emptyset\}} \left( \sqrt{\tilde{P}_{r_{\text{out}}}^{\text{rat}}(z|x, a, b)} - \sqrt{\tilde{P}_{r^\star}^{\text{rat}}(z|x, a, b)} \right)^2$$

$$= \int_{z \in \mathcal{Z}\backslash\{\emptyset\}} \left( \sqrt{p_{\text{obs,rat}} P_{r_{\text{out}}}^{\text{rat}}(z|x, a, b)} - \sqrt{p_{\text{obs,rat}} P_{r^\star}^{\text{rat}}(z|x, a, b)} \right)^2$$

$$= p_{\text{obs,rat}} \int_{z \in \mathcal{Z}\backslash\{\emptyset\}} \left( \sqrt{P_{r_{\text{out}}}^{\text{rat}}(z|x, a, b)} - \sqrt{P_{r^\star}^{\text{rat}}(z|x, a, b)} \right)^2$$

$$= p_{\text{obs,rat}} D_H^2(P_{r_{\text{out}}}^{\text{rat}}(\cdot|x, a, b), P_{r^\star}^{\text{rat}}(\cdot|x, a, b)).$$

Analogously, one can get

$$D_H^2(\tilde{P}_{r_{\text{out}}}^{\text{rank}}(\cdot|x, a, b), \tilde{P}_{r^\star}^{\text{rank}}(\cdot|x, a, b)) = p_{\text{obs,rank}} D_H^2(P_{r_{\text{out}}}^{\text{rank}}(\cdot|x, a, b), P_{r^\star}^{\text{rank}}(\cdot|x, a, b)).$$

At this point, with the same steps followed for the proof of Theorem 4.2, we can get

$$D_H^2(P_{r_{\text{out}}}^{\text{rank}}(\cdot|x, a, b), P_{r^\star}^{\text{rank}}(\cdot|x, a, b)) \geq \frac{(r_{\text{out}}(x, a) - r_{\text{out}}(x, b) - r^\star(x, a) - r^\star(x, b))^2}{16e^{4R_{\max}}R_{\max}^2}$$

and

$$D_H^2(P_{r_{\text{out}}}^{\text{rat}}(\cdot|x, a, b), P_{r^\star}^{\text{rat}}(\cdot|x, a, b)) \geq \frac{(r_{\text{out}}(x, a) - r_{\text{out}}(x, b) - r^\star(x, a) - r^\star(x, b))^2}{2(R_{\max} - R_{\min})^2 + 4\mathbb{V}}.$$

All in all, we get,

$$(1-\alpha)D_H^2(\tilde{P}_{r_{\text{out}}}^{\text{rank}}(\cdot|x,a,b), \tilde{P}_{r^\star}^{\text{rank}}(\cdot|x,a,b)) + \alpha D_H^2(\tilde{P}_{r_{\text{out}}}^{\text{rat}}(\cdot|x,a,b), \tilde{P}_{r^\star}^{\text{rat}}(\cdot|x,a,b))$$

$$\geq (1-\alpha)\frac{p_{\text{obs,rank}}(r_{\text{out}}(x,a) - r_{\text{out}}(x,b) - r^\star(x,a) - r^\star(x,b))^2}{16e^{4R_{\max}}R_{\max}^2}$$

$$+ \alpha\frac{p_{\text{obs,rat}}(r_{\text{out}}(x,a) - r_{\text{out}}(x,b) - r^\star(x,a) - r^\star(x,b))^2}{2(R_{\max} - R_{\min})^2 + 4\mathbb{V}}.$$

Therefore,

$$\left(\frac{(1-\alpha)p_{\text{obs,rank}}}{16e^{4R_{\max}}R_{\max}^2} + \frac{\alpha p_{\text{obs,rat}}}{2(R_{\max} - R_{\min})^2 + 4\mathbb{V}}\right)\text{Error}_{\pi_{\text{ref}}}(\pi_{\text{out}}) \leq \frac{2\log(|\Pi|/\delta)}{N},$$

which leads to the following bound with probability $1-\delta$,

$$\text{Error}_{\pi_{\text{ref}}}(\pi_{\text{out}}) \leq \left(\frac{16e^{4R_{\max}}R_{\max}^2(2(R_{\max} - R_{\min})^2 + 4\mathbb{V})/p_{\text{obs,rank}}p_{\text{obs,rat}}}{(2(R_{\max}-R_{\min})^2+4\mathbb{V})/p_{\text{obs,rat}}(1-\alpha) + 16e^{4R_{\max}}R_{\max}^2/p_{\text{obs,rank}}\alpha}\right)\frac{2\log(|\Pi|/\delta)}{N}.$$

Therefore, for $\alpha = \mathbb{1}\left\{16e^{4R_{\max}}R_{\max}^2/p_{\text{obs,rank}} \geq (2(R_{\max}-R_{\min})^2+4\mathbb{V})/p_{\text{obs,rat}}\right\}$, we obtain

$$\text{Error}_{\pi_{\text{ref}}}(\pi_{\text{out}}) \leq \min\left\{\frac{(2(R_{\max} - R_{\min})^2 + 4\mathbb{V})}{p_{\text{obs,rat}}}, \frac{16e^{4R_{\max}}R_{\max}^2}{p_{\text{obs,rank}}}\right\}\frac{2\log(|\Pi|/\delta)}{N}.$$

which allows to conclude the proof. The equality in the theorem statement holds because $p_{\text{obs,rat}}N = \mathbb{E}[N_{\text{rat}}]$ and $p_{\text{obs,rank}}N = \mathbb{E}[N_{\text{rank}}]$. $\qquad\square$

# E    TECHNICAL LEMMAS

This section contains technical results that are used in proving the guarantees for RDPO and ML-RDPO. We start with an important duality result between RLHF and DPO. Before delving in the proof, let us state Assumption 1 in the following equivalent form.

**Assumption 2.** *Let us consider any reward* $r : \mathcal{X} \times \mathcal{A} \to [R_{\min}, R_{\max}]$. *Then, we assume that* $\Pi$ *is such that for any* $\gamma \in [0,1]$, *it holds that*

$$\underset{\pi\in\Pi}{\arg\max}\ \langle\pi, r\rangle - \beta\gamma D_{\chi^2}(\pi, \pi_{\text{ref}}) - \beta D_{\text{KL}}(\pi, \pi_{\text{ref}})$$

$$= \underset{\pi\in\Pi_{\text{all}}}{\arg\max}\ \langle\pi, r\rangle - \beta\gamma D_{\chi^2}(\pi, \pi_{\text{ref}}) - \beta D_{\text{KL}}(\pi, \pi_{\text{ref}})$$

The assumption says that the policy class $\Pi$ should be expressive enough to realize the solution of all RLHF problems associated with all possible bounded reward functions. The next lemma leverages the rewriting of Assumption 1 to show that any policy $\pi$ is the maximizer of the regularized RL problem having as reward the function $\beta\phi(\pi/\pi_{\text{ref}})$.

**Lemma E.1.** *Let Assumption 2 hold. For any policy* $\pi \in \Pi$, *let us consider the reward function* $r_\pi(x,a) = \beta\phi\left(\frac{\pi(a|x)}{\pi_{\text{ref}}(a|x)}\right)$ *where* $\phi(x) = \gamma x + \log(x)$. *Then, if* $R_{\min} \leq r_\pi(x,a) \leq R_{\max}$ *for all* $x, a \in \mathcal{X} \times \mathcal{A}$, *it holds true that*

$$\pi \in \underset{p\in\Pi}{\arg\max}\ \langle p, r_\pi\rangle - \beta\gamma D_{\chi^2}(p, \pi_{\text{ref}}) - \beta D_{\text{KL}}(p, \pi_{\text{ref}}).$$

*Proof.* By Assumption 2, we have that $\pi$ is the solution when the domain is enlarged from $\Pi$ to $\Pi_{\text{all}}$, that is

$$\pi \in \underset{p\in\Pi_{\text{all}}}{\arg\max}\ \langle p, r_\pi\rangle - \beta\gamma D_{\chi^2}(p, \pi_{\text{ref}}) - \beta D_{\text{KL}}(p, \pi_{\text{ref}})$$

At this point, noticing that $\gamma D_{\chi^2}(\pi, \pi') + D_{\text{KL}}(\pi, \pi')$ can be interpreted as $f$-divergence generated by $f(x) = \gamma\frac{x^2}{2} + x\log(x)$. Notice that $f'(x) = \gamma x + \log(x) + 1$ and therefore since $0 \notin \text{dom}(f')$. Therefore, the conditions of (Huang et al., 2024, Lemma F.2) are satisfied and we can invoke their result to conclude the proof. $\qquad\square$

Next, we present two results that bound the generalization error for the estimator learned via ranking information. The first result upper bounds the absolute value error of $r_{\text{out}}$ in terms of the Hellinger divergences between the Bradley-Terry model with ground truth reward $r$ and the one with reward $r_{\text{out}}$.

**Lemma E.2.** *(adapted from (Huang et al., 2024, Lemma F.5)) Recall that $r^\star(x,a) - r^\star(x,b) \in [R_{\min} - R_{\max}, R_{\max} - R_{\min}]$ and $r_{\text{out}}(x,a) - r_{\text{out}}(x,b) \in [R_{\min} - R_{\max}, R_{\max} - R_{\min}]$ , then we define*

$$P_r(\mathbb{1}\{a \text{ is preferred to } b\} \mid x,a,b) = \sigma(r(x,a) - r(x,b))$$

*for any $r \in \mathcal{R}_{\Pi'}$. Then, it holds that*

$$|r^\star(x,a) - r^\star(x,b) + r_{\text{out}}(x,a) - r_{\text{out}}(x,b)|$$

$$\leq 4e^{2R_{\max}} R_{\max} \sqrt{\sum_{z \in \{0,1\}} \left(\sqrt{P_{r^\star}(z|x,a,b)} - \sqrt{P_{r_{\text{out}}}(z|x,a,b)}\right)^2}$$

Second, we invoke a classic result on the generalization error of the maximum likelihood estimator to bound the Hellinger divergence.

**Lemma E.3.** *(Conditional MLE Risk) For any policy class $\Pi$, let us consider the following policy-induced reward class*

$$\mathcal{R}_\Pi = \left\{ r \mid \exists \pi \in \Pi \ s.t. \ r(x,a) = \beta \phi\left(\frac{\pi(a|x)}{\pi_{\text{ref}}(a|x)}\right) \ \forall x, a \in \mathcal{X} \times \mathcal{A}\right\}.$$

*Moreover, consider the conditional density $P_r : \mathcal{X} \times \mathcal{A} \times \mathcal{A} \to [0,1]$, the dataset $\mathcal{D}_{\text{rank}} = \{x_i, a_i^+, a_i^-\}$ with $|\mathcal{D}_{\text{rank}}| = N_{\text{rank}}$ and the estimator*

$$r_{\text{out}} = \operatorname*{argmax}_{r \in \mathcal{R}_\Pi} \sum_{i=1}^{N_{\text{rank}}} \log P_r(a_i^+ \text{ is preferred to } a_i^- | x_i, a_i^-, a_i^+).$$

*Then, it holds that with probability $1 - \delta$*

$$\sum_{x \in \mathcal{X}} \nu_0(x) \sum_{a,b \in \mathcal{A} \times \mathcal{A}} \pi_{\text{ref}}(a|x)\pi_{\text{ref}}(b|x) \sum_{z \in \{0,1\}} \left(\sqrt{P_{r_{\text{out}}}(z|x,a,b)} - \sqrt{P_{r^\star}(z|x,a,b)}\right)^2$$

$$\leq \frac{2\log(|\Pi|/\delta)}{N_{\text{rank}}}.$$

Finally, we provide the bound for the least square estimate using as targets the observed rating gaps, which we assume to be unbiased and Gaussian distributed.

**Lemma E.4.** *(Concentration for the rating reward) Let us consider the generalization error for the rating reward estimator*

$$\text{Error}_{\pi_{\text{ref}}}(\hat{r}_{\text{rat}}) = \sum_{x \in \mathcal{X}} \nu_0(x) \sum_{a \in \mathcal{A}} \pi_{\text{ref}}(a|x)\pi_{\text{ref}}(b|x) \left(r^\star(x,a) - r^\star(x,b) - \hat{r}_{\text{rat}}(x,a) + \hat{r}_{\text{rat}}(x,b)\right)^2$$

*where*

$$\hat{r}_{\text{rat}} = \operatorname*{argmin}_{r \in \mathcal{R}} \sum_{i=1}^{N_{\text{rat}}} (r(x_i', \bar{a}_i) - r(x_i', \tilde{a}_i) - \Delta_{\hat{r}}^i)^2,$$

*and $\Delta_{\hat{r}}^i | x_i', \bar{a}_i, \tilde{a}_i$ is a $\mathbb{V}$-sub Gaussian random variable with mean $r^\star(x_i', \bar{a}_i) - r^\star(x_i', \tilde{a}_i)$. Then we have that with probability at least $1 - 2\delta$,*

$$\text{Error}_{\pi_{\text{ref}}}(\hat{r}_{\text{rat}}) \leq \mathcal{O}\left(\frac{(R_{\max} - R_{\min} + \sqrt{\mathbb{V}\log(2N_{\text{rat}}/\delta)})^2 \log(\frac{|\mathcal{R}|}{\delta})}{N_{\text{rat}}}\right)$$

*Proof.* By the definition of a subgaussian random variable, we have that

$$\mathbb{P}\left(\left|\Delta_{\hat{r}}^i - r^\star(x_i', \bar{a}_i) + r^\star(x_i', \tilde{a}_i)\right| \geq t \ \middle| \ x_i', \bar{a}_i, \tilde{a}_i\right) \leq 2e^{-t^2/\mathbb{V}}$$

for any scalar $t \in \mathbb{R}$. Therefore, for $t = \sqrt{\mathbb{V} \log (2N_{\text{rat}}/\delta)}$, and a union bound we have that

$$\mathbb{P}\left( \left| \Delta_{\hat{r}}^i - r^\star(x_i', \bar{a}_i) + r^\star(x_i', \tilde{a}_i) \right| \geq \sqrt{\mathbb{V} \log (2N_{\text{rat}}/\delta)} \;\; \forall i \in [N_{\text{rat}}] \;\; \middle| \;\; x_i', \bar{a}_i, \tilde{a}_i \right) \leq \delta$$

$$\implies \mathbb{P}\left( \left| \Delta_{\hat{r}}^i \right| \geq R_{\max} - R_{\min} + \sqrt{\mathbb{V} \log (2N_{\text{rat}}/\delta)} \;\; \forall i \in [N_{\text{rat}}] \;\; \middle| \;\; x_i', \bar{a}_i, \tilde{a}_i \right) \leq \delta$$

Therefore, with probability $1 - \delta$ the regression targets $\left\{ \Delta_{\hat{r}}^i \right\}_{i=1}^{N_{\text{rat}}}$ are in the interval $[-R_{\max} + R_{\min} - \sqrt{\mathbb{V} \log (2N_{\text{rat}}/\delta)}, R_{\max} - R_{\min} + \sqrt{\mathbb{V} \log (2N_{\text{rat}}/\delta)}]$. Invoking Foster & Rakhlin (2023)[Chapter 1, Exercise 1 ], we have that under the event for $\mathcal{E}_{\text{bounded}} = \left\{ \Delta_{\hat{r}}^i \in [-B, B], B = R_{\max} - R_{\min} + \sqrt{\mathbb{V} \log (2N_{\text{rat}}/\delta)} \right\}$, it holds that

$$\mathbb{P}\left( \text{Error}_{\pi_{\text{ref}}}(\hat{r}_{\text{rat}}) \leq \mathcal{O}\left( \frac{B^2 \log(\frac{|\mathcal{R}|}{\delta})}{N_{\text{rat}}} \right) \middle| \mathcal{E}_{\text{bounded}} \right) \geq 1 - \delta.$$

Finally, to remove the conditioning $\mathcal{E}_{\text{bounded}}$, by the law of total probability we have that

$$\mathbb{P}\left( \text{Error}_{\pi_{\text{ref}}}(\hat{r}_{\text{rat}}) \geq \mathcal{O}\left( \frac{B^2 \log(\frac{|\mathcal{R}|}{\delta})}{N_{\text{rat}}} \right) \right)$$

$$= \mathbb{P}\left( \text{Error}_{\pi_{\text{ref}}}(\hat{r}_{\text{rat}}) \geq \mathcal{O}\left( \frac{B^2 \log(\frac{|\mathcal{R}|}{\delta})}{N_{\text{rat}}} \right) \middle| \mathcal{E}_{\text{bounded}} \right) \mathbb{P}\left( \mathcal{E}_{\text{bounded}} \right)$$

$$+ \mathbb{P}\left( \text{Error}_{\pi_{\text{ref}}}(\hat{r}_{\text{rat}}) \geq \mathcal{O}\left( \frac{B^2 \log(\frac{|\mathcal{R}|}{\delta})}{N_{\text{rat}}} \right) \middle| \mathcal{E}_{\text{bounded}}^c \right) \mathbb{P}\left( \mathcal{E}_{\text{bounded}}^c \right)$$

$$\leq \mathbb{P}\left( \text{Error}_{\pi_{\text{ref}}}(\hat{r}_{\text{rat}}) \geq \mathcal{O}\left( \frac{B^2 \log(\frac{|\mathcal{R}|}{\delta})}{N_{\text{rat}}} \right) \middle| \mathcal{E}_{\text{bounded}} \right) + \mathbb{P}\left( \mathcal{E}_{\text{bounded}}^c \right)$$

$$\leq 2\delta.$$

$\square$

# F    ADDITIONAL EXPERIMENTS

In this section we report the win rates between some model pairs in the 7/8 B parameters model in Appendix F.1 experiments on the smaller models SmolLM2-135M[9], SmoLM2-360M [10] and SmolLM2-1.7B [11] in Appendix F.3. Then, we report additional experiments with perturbed rating information in Appendix F.2.

## F.1    RELATIVE WIN RATES AT 7/8B SCALE

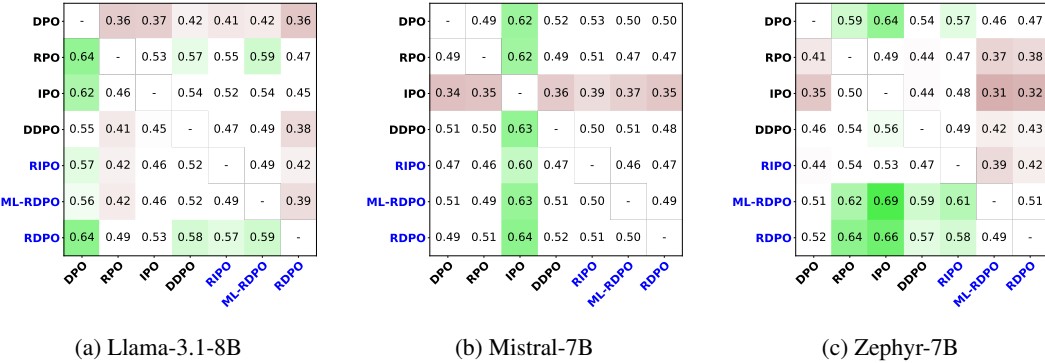

(a) Llama-3.1-8B                    (b) Mistral-7B                    (c) Zephyr-7B

Figure 6: Model by model comparison using Llama, Mistral and Zephyr as base models. Each entry of the table reports the win rate of the row player against the column player.

We see that RDPO achieves a high win rate against any other alignment methods for all three choices for $\pi_{\text{ref}}$. ML-RDPO generally achieves an even higher win rate for Zephyr and Mistral while it underperforms slightly in Llama. IPO and RPO are strong baselines for the experiment with Llama as a base model. The attentive reader might notice that the antidiagonal elements do not sum to one. This is because we run two independent judgments for each pair of models.

## F.2    ADDITIONAL EXPERIMENTS WITH NOISY RATING INFORMATION

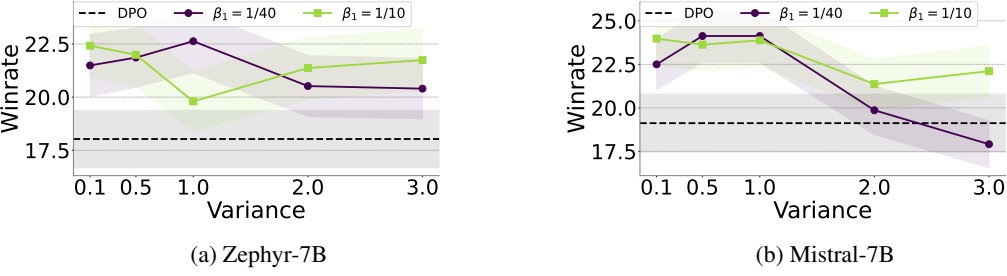

(a) Zephyr-7B                          (b) Mistral-7B

Figure 7: Variance ablation in Mistral-7B and Zephyr-7B.

In Figure 7 we report the win rate against GPT-4 as judged by Claude-Sonnet-3.5v2 for the base models Zephyr-7B and Mistral-7B. The experiment aims at monitoring the win rate as a function of the variance of the Gaussian noise injected in the `ultrafeedback` rating.

As we found using Llama-8B as base model ( see Figure 3b), the choices for $\beta_1 = 1/10$ and $\beta_1 = 1/40$ perform similarly at low variance level while $\beta_1 = 1/10$ performs better when high variance values are considered. This behaviour is consistent with the theoretical prediction that a higher $\beta_1$ should be considered when the rating information is inaccurate.

---

[9] https://huggingface.co/HuggingFaceTB/SmolLM2-135M
[10] https://huggingface.co/HuggingFaceTB/SmolLM2-360M
[11] https://huggingface.co/HuggingFaceTB/SmolLM2-1.7B

### F.3 Experiments with Smol-LM2 Family

We experiment with the SmolLM2 family as an initial test; however, small language models can be important in some situations, such as on-device applications Xu et al. (2024a). Therefore, we include the results in the following. In particular, we report the results of the relative win rates between any pair of considered alignment methods. We observe that for SmolLM2 models, it is possible to

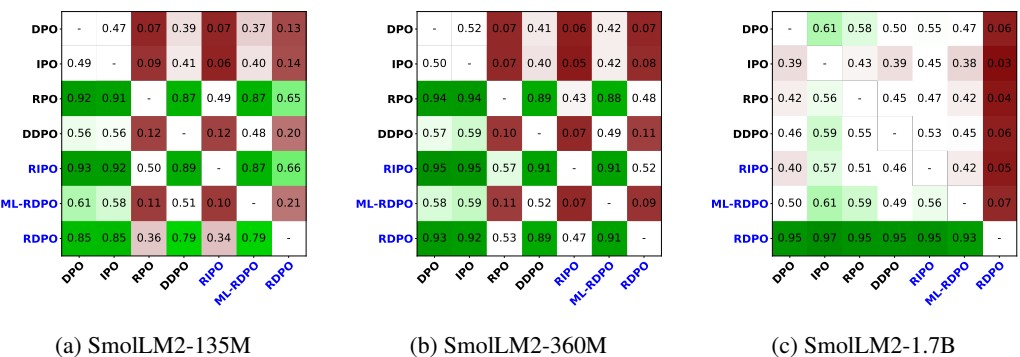

| (a) SmolLM2-135M | (b) SmolLM2-360M | (c) SmolLM2-1.7B |

Figure 8: Model by model comparison using Llama, Mistral and Zephyr as base model.

achieve larger win rates, and in this case, the benefit of rating is even more evident than at 7B/8B scale. At 135M, the best methods are RPO Adler et al. (2024a) and our RIPO, while RDPO is the third best performing in this setting. These three algorithms are comparable at 360M scale, while RDPO outperforms all the other methods neatly at 1.7B scale.

ML-RDPO does not perform as well in this setting, but it is always improving over both DPO and Distilled DPO as predicted by Theorem 4.2.

In these tables, we compare the "best" epochs for each algorithm. Best epoch here means the epoch that achieved the highest win rate against $\pi_{\mathrm{ref}}$, which is reported in Figure 9.

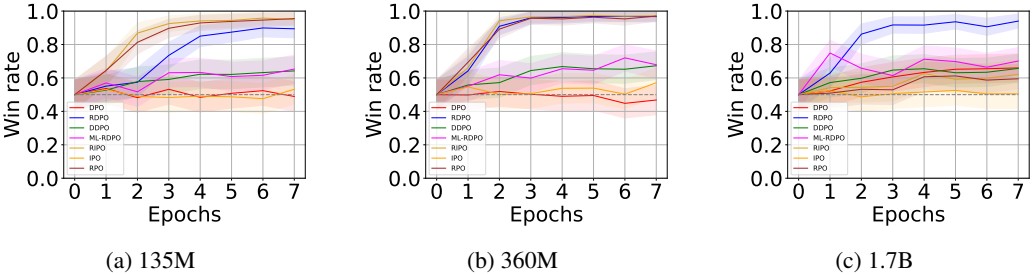

| (a) 135M | (b) 360M | (c) 1.7B |

Figure 9: Win rate against $\pi_{\mathrm{ref}}$ for the SmolLM2 family experiments.

### F.4 Ablation for $\beta_1$ and $\mathbb{V}$

For deciding the hyperparameters to use in the experiments shown in the main text, we run ML-RDPO with 5 possible choices for $\mathbb{V}$ and RDPO with 5 possible choices for $\beta_1$ using Zephyr-7B as base model. We select the configuration with the best-performing win rate against $\pi_{\mathrm{ref}}$ for the other experiments. The best hyperparameters found in this way are $\beta_1 = 1/10$ for RDPO and $\mathbb{V} = 1/100$ for ML-RDPO. We report the win rates achieved by the configuration we tried in Table 1

For both algorithms, we used otherwise standard DPO choices, that is $\beta = 0.1$ and learning rate $1e - 6$.

Surprisingly, the best hyperparameters for the SmolLM2 family are different. In particular, in Figure 10 we show that for the setting $\pi_{\mathrm{ref}} = \mathtt{SmolLM2-1.7B}$ the best value for $\beta_1$ is 0.005, which is much smaller than the values found at 7B/8B level. Recall that a smaller $\beta_1$ translates to higher

Table 1: Zephyr-7B, Ablation for win rate against $\pi_{\mathrm{ref}}$ on Alpaca Eval

| Model | win rate | Standard Error |
|---|---|---|
| ML-RDPO $\mathbb{V} = 1$ | 2.43 | 0.539 |
| ML-RDPO $\mathbb{V} = 1/10$ | 17.68 | 1.398 |
| ML-RDPO $\mathbb{V} = 1/50$ | 19.07 | 1.384 |
| ML-RDPO $\mathbb{V} = 1/100$ | **23.45** | 1.494 |
| ML-RDPO $\mathbb{V} = 1/200$ | 19.59 | 1.399 |
| RDPO $\beta_1 = 1$ | 11.68 | 1.133 |
| RDPO $\beta_1 = 1/10$ | **23.38** | 1.491 |
| RDPO $\beta_1 = 1/40$ | 17.83 | 1.352 |
| RDPO $\beta_1 = 1/100$ | 16.04 | 1.295 |
| RDPO $\beta_1 = 1/125$ | 15.17 | 1.266 |
| Zephyr-7B-SFT-full | 15.11 | 1.262 |

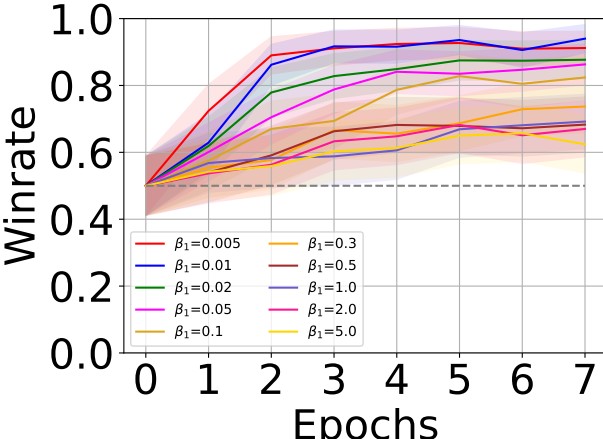

Figure 10: Win rate vs base model SmolLM2-1.7B in Alpaca-Eval for different values of $\beta_1$ used in Algorithm 1

trust for the ratings. Interestingly, it seems that these experiments suggest that the model size plays a role in choosing the right level of trust. This fact is not captured by our current theory, which neglects the optimization aspects of the problem. We believe it is an interesting open question to develop a deeper understanding of this phenomenon, either empirically or theoretically.

## F.5 ON THE RESTRICTION TO $\Pi'$ IN THEOREM 4.1

In this section, we investigate a practical implementation of RDPO which embodies some penalty to ensure that the output policy is in the restricted class $\Pi'$ which we recall being the subset of $\Pi$ such that $\pi \in \Pi'$ implies that $\left| \Delta_{\log \pi / \pi'_{\mathrm{ref}}}(x, a, b) \right| \leq \Delta_{\max}$ where $\Delta_{\max} := R_{\max} - R_{\min}$. To this end, we consider the following minimization problem

$$\operatorname*{argmin}_{\pi \in \Pi} \sum_{i=1}^{N} - \log \sigma \left( \beta \Delta^i_{\log \pi / \pi_{\mathrm{ref}}} - \frac{\beta}{\beta_1} \Delta^i_{\hat{r}} \right) +$$

$$+ \lambda_1 \left( \Delta^i_{\log \pi / \pi_{\mathrm{ref}}} - \frac{\Delta^i_{\hat{r}}}{\beta_1} - \Delta_{\max} \right) \mathbb{1} \left\{ \Delta^i_{\log \pi / \pi_{\mathrm{ref}}} - \frac{\Delta^i_{\hat{r}}}{\beta_1} \geq \Delta_{\max} \right\}$$

$$+ \lambda_2 \left( -\Delta^i_{\log \pi / \pi_{\mathrm{ref}}} + \frac{\Delta^i_{\hat{r}}}{\beta_1} - \Delta_{\max} \right) \mathbb{1} \left\{ \Delta^i_{\log \pi / \pi_{\mathrm{ref}}} - \frac{\Delta^i_{\hat{r}}}{\beta_1} \leq -\Delta_{\max} \right\}$$

where $\lambda_1, \lambda_2 \in \mathbb{R}^+$ are penalties hyperparameters. The larger $\lambda_1, \lambda_2$ are, the smallest the constraint violation of $\pi_{\mathrm{out}}$ is.

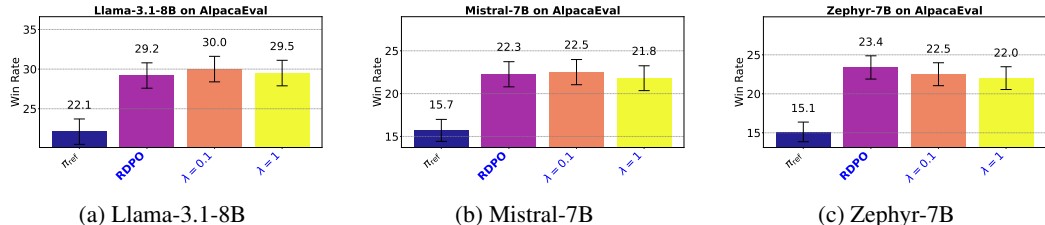

| (a) Llama-3.1-8B | (b) Mistral-7B | (c) Zephyr-7B |

Figure 11: Comparison between RDPO and RDPO with penalty parameters $\lambda_1, \lambda_2$ equals to the value of $\lambda$ reported in the plots.

As shown in Figure 11, we observe an improvement for Llama and Mistral as base model while the performance without penalty is better for $\pi_{\text{ref}}$ chosen to be Zephyr-7B. However, in all cases the difference between RDPO with or without penalties is not remarkable, therefore, while the restriction to $\Pi'$ is needed for the theoretical result in Theorem 4.1, we consider it as optional in practice. In particular, it might be avoided when handling a simpler objective is desirable.

