# OpenReview forum: "Alignment from Ranking and Rating Information"
_ICLR.cc/2026/Conference — Submitted to ICLR 2026_

### Official Review · Reviewer_zdGC · 2025-10-22

**Soundness:** 3
**Presentation:** 3
**Contribution:** 2
**Rating:** 2
**Confidence:** 3

**Summary:**

This paper identifies a key limitation in Direct Preference Optimization (DPO): by using only binary pairwise preferences, DPO ignores the magnitude of the preference. The authors argue this ambiguity—where DPO cannot distinguish between a pair of high-quality responses and a pair with one good and one bad response —can incentivize policies that favor out-of-distribution responses, a phenomenon termed "likelihood displacement".

To address this, the paper proposes to augment the training data with "rating gap" information, which specifies *how much* a preferred response is better than a rejected one. The authors present three new algorithms that leverage this information:
1.  **Rating DPO (RDPO)**
2.  **Rating IPO (RIPO)**
3.  **Maximum-Likelihood-based RDPO (ML-RDPO)**

The paper provides theoretical guarantees, suggesting these methods can achieve faster statistical rates ("acceleration") when the rating gap information is accurate and maintain performance even when this information is noisy, provided the hyperparameters are tuned correctly. Empirically, the authors demonstrate that RDPO and ML-RDPO outperform DPO and other DPO-style algorithms on the AlpacaEval and ArenaHard benchmarks across several base models.

**Strengths:**

1. **Well-Motivated Problem:** The paper's core premise is intuitive and sound. DPO's reliance on simple binary preferences clearly discards a rich source of information about response quality. The idea of incorporating the *magnitude* of this preference via rating gaps is a logical and compelling extension.
2. **Principled Algorithm Derivation:** The authors provide principled, step-by-step derivations for their proposed algorithms. RDPO and RIPO are derived by modifying the RLHF objective to include rating information, while ML-RDPO is derived from a maximum-likelihood perspective .
3. **Theoretical Analysis:** The paper includes a theoretical analysis that formalizes the concepts of "acceleration"  and "robustness", providing a useful (if idealized) framework for understanding *why* these methods should be beneficial.
4. **Strong Empirical Performance:** The experimental results in Figures 1 and 2 are solid, showing that RDPO and ML-RDPO consistently achieve higher win rates than DPO and other baselines on the chosen benchmarks.

**Weaknesses:**

1. **Motivation-Experiment Mismatch:** The paper's primary motivation is to solve the "likelihood displacement" problem, which it claims incentivizes OOD responses. However, the experimental evaluation never measures this phenomenon. The experiments are limited to win-rate comparisons on standard benchmarks. While the method improves win rates, there is no evidence provided to substantiate the claim that it actually mitigates likelihood displacement or improves OOD generalization.
2. **Impractical Theoretical Assumptions:** The theoretical guarantees rely on assumptions that are unlikely to hold in practice.
    * Assumption 1  is particularly strong, as it requires the policy class $\Pi$ (i.e., the LLM) to be expressive enough to contain the optimal closed-form policy $\pi_{\beta}^{*}$ for *any* bounded reward function. The paper does not justify why this would be the case for real-world models.
    * ML-RDPO relies on simplifying assumptions for its derivation, such as the conditional independence between the preference label ($z_i$) and the rating gap ($\Delta_r^i$).
3. **Hyperparameter Tuning and Practicality:** The method's practical utility is a major concern. The theoretical analysis itself provides guidance for setting the crucial hyperparameters $\beta$ and $\beta_1$ based on the ratio $Err_{DPO}(N,\delta)/Err_{\pi_{ref}}(\hat{r})$ . This ratio is fundamentally unknowable in a practical setting, as it requires access to the (unknown) true reward error. The robustness experiments in Figure 3  confirm that performance is highly sensitive to finding the *correct* $\beta_1$ (trust in ratings) , but the paper offers no practical heuristic for setting this value, likely requiring expensive, dataset-specific tuning.
4. **Limited Experimental Scope:** The empirical validation is narrow. All main experiments are conducted using only the `ultrafeedback_binarized` dataset and evaluated on only two benchmarks, `AlpacaEval` and `ArenaHard`. This makes it difficult to know if the observed performance gains are broadly generalizable or an artifact of the specific properties of this dataset.

**Questions:**

See the weaknesses. Most concerns arise from the disconnection between motivation and evaluation, as well as the practical utility.

---

> ### Author Response · Authors · 2025-11-15
>
> Thanks for your review. We respectfully think that there are several misunderstandings of our paper in your review. Please read our clarifications below. We hope that our answers will solve your misunderstandings and lead to a positive evaluation of our work.
>
>
> ***The paper's primary motivation is to solve the "likelihood displacement" problem, which it claims incentivizes OOD responses. However, the experimental evaluation never measures this phenomenon. The experiments are limited to win-rate comparisons on standard benchmarks. While the method improves win rates, there is no evidence provided to substantiate the claim that it actually mitigates likelihood displacement or improves OOD generalization.***
>
> We never said that the goal is to solve likelihood displacement or OOD generalization.  Our goal is to build an effective and principled framework to learn jointly from preferences and rating information.
> We touch upon the "likelihood displacement" problem in the introduction only to explain that ratings might be useful to assess whether the winner losing loglikelihood has good or low ratings.
> Notice, however, that the likelihood displacement per se is not a problem but just a phenomenon.
>
> Moreover, we never mention OOD generalization in our work, so we are uncertain about the point raised by the reviewer. We are happy to discuss more if the reviewer could clarify what they mean by OOD generalization in the context of alignment.
>
> ***The theoretical guarantees rely on assumptions that are unlikely to hold in practice.
> Assumption 1 is particularly strong, as it requires the policy class
>  (i.e., the LLM) to be expressive enough to contain the optimal closed-form policy  for any bounded reward function. The paper does not justify why this would be the case for real-world models.
> ML-RDPO relies on simplifying assumptions for its derivation, such as the conditional independence between the preference label and the rating gap.***
>
> Notice that without assuming that the policy class realizes the optimal closed-form policy is not even possible to derive DPO. Indeed, if this assumption does not hold, the solution set of the DPO loss might be totally disjoint from the one of the RLHF problem.
>
> Moreover, Assumption 1 is commonly used in LLM theory (see e.g. Assumption 3.1 in [1,2]).
>
> [1] Huang et al., "Correcting the Mythos of KL-Regularization", https://arxiv.org/pdf/2407.13399
>
> [2] Xie et al., "Exploratory Preference Optimization", https://arxiv.org/pdf/2405.21046
>
>
> Finally, we also think that the conditional independence is not a strong assumption. Indeed, it can be verified in practice by obtaining the preference and the rating with two separate queries. Moreover, the assumption is not needed by RDPO but only by ML-RDPO.
>
> ***The method's practical .... but the paper offers no practical heuristic for setting this value, likely requiring expensive, dataset-specific tuning.***
>
> We respectfully disagree with the reviewer on this point.
>
> Notice that our RDPO comes with only one parameter to tune, i.e. $\beta_1$ and its choice seems to be quite robust both across models of the same size and across noise levels (see Figure 3).
>
> It is true that we need unknown quantities to set $\beta_1$. However, we remark that our work is the first to show the important acceleration robustness tradeoff. Making the $\beta_1$ adaptive can be the focus of future work.
>
> Finally, notice that tuning $\beta_1$ was not difficult. We just tried 5 different values on Zephyr-7B and Alpaca-Eval as an evaluation baseline and kept exactly the same value in all other experiments.
> Moreover, we disagree that Figure 3 shows a fragility of the parameters $\beta_1$ indeed for the value chosen via our ablation in Appendix F.4, i.e. $\beta_1 = 1/10$, we obtain solid performance for all levels of noise.
> In conclusion, the tuning was neither expensive nor dataset or model specific.
>
> ***The empirical validation is narrow. All main experiments are conducted using only the ultrafeedback_binarized dataset and evaluated on only two benchmarks, AlpacaEval and ArenaHard. This makes it difficult to know if the observed performance gains are broadly generalizable or an artifact of the specific properties of this dataset.***
>
> We used ultrafeedback as training dataset because it is the only commonly used training set that contains both ranking and rating information. Moreover, ultrafeedback is a very commonly used training set and evaluation benchmark. For example, it has been used in the SimPO papers [1].
>
> Similarly, the evaluation datasets AlpacaEval and ArenaHard are the ones commonly used in the literature. Could the reviewer refer to which particular evaluation benchmark we are missing?
>
> [1] Meng et al. "SimPO: Simple Preference Optimization with a Reference-Free Reward"

---

> > ### Author Response · Authors · 2025-11-15
> >
> > In conclusion, we respectfully disagree with the concerns that the reviewer brought up:
> >
> > (i) We never claimed to address likelihood displacements and OOD generalization; therefore, we do not see the need for providing experiments to investigate these phenomena.
> >
> > (ii) Our methods are not difficult to tune as they require the tuning of a single hyperparameter ($\beta_1$ for RDPO and $\mathbb{V}$ for ML_RDPO )
> >
> > (iii) The evaluation is comparable to those of recent works in the same area.
> >
> > We remain available to discuss more if you would like to better clarify your concerns regarding the submission. Otherwise,
> > we would appreciate if, in light of our responses, you could reconsider your score.

---

> > > ### Comment · Reviewer_zdGC · 2025-11-20
> > >
> > > Thank the authors for your prompt response. For your replies:
> > >
> > > 1. The motivation problem. In the first paragraph of introduction, the authors describes the potential likelihood displacement problem caused by the lack of some kind of absolute quality. And here an example of abnormally favored OOD responses is given. I must apologize that this is kind of distracting so I misinterpreted this as some part of your motivation.
> > > 2. You are correct on your point about assumption 1. Actually I was puzzled by the later Theorem 4.1, as it requires strict boundedness to ensure the concentration inequalities (variance bounds) hold. I was thinking if the class $\Pi$ is large enough to satisfy Assumption 1, it is not guaranteed that the optimal policy $\pi^*$ will fall inside the bounded region $\Pi'$ mandated by Theorem 4.1 when the latent reward $r^*$ takes extreme values. But I figured this out in Appendix F.5 and you are generally right.
> > > 3. For the tuning of $\beta_1$. I think my view of "fragile" and your view of "solid, neither expensive nor model/dataset specific" are just two sides of the same coin. In your section 4.1 you mention that the ratio of $\beta / \beta_1$ should be coherent with the confidence level with rating over ranking. But this trust/confidence thing is just qualitative instead of quantitative. The noise experiment results are inspiring showing your insights, but that is it. In reality we still cannot know how to tune $\beta_1$ with a new dataset, and how much it would shift, given the fact you only use one training set `ultrafeedback`.  I think you cannot even specify what is a safe choice of $\beta_1$ to make your RDPO to perform at least as well as DPO, for a completely new setup in practice. It would greatly improve the application value if you could provide more insights on this point.
> > > 4. For the benchmark selection. The point is that, both benchmarks you chose, the `AlpacaEval` and `ArenaHard`, are relative comparison benchmarks without a golden standard. Of course it is a common practice to use such benchmarks in RL evaluation, but I think it would be more persuasive to consider some benchmarks with reference answers. For example, try `IFEval` to see the general intruction-following capabilities.
> > >
> > > In all, I sincerely apologize for my careless mistakes in your theory analysis part. But I think the tuning of $\beta_1$ still lacks some kind of practical instructions to operate - at least current experiment settings do not ensure the transferability, from my perspective. Also, while `ultrafeedback` is a general-purposed dataset, evaluating on benchmarks with golden standards would help support your empirical study better. I think `IFEval` is a quite efficient option to test on. I will increase my rating to 6, with your theoretical concerns addressed.

---

> > > > ### Author Response · Authors · 2025-11-21
> > > >
> > > > Dear reviewer,
> > > >
> > > > Thanks a lot for reconsidering your evaluation in light of our responses !
> > > >
> > > > Best,
> > > > Authors

---

### Official Review · Reviewer_2n9a · 2025-10-31

**Soundness:** 3
**Presentation:** 3
**Contribution:** 3
**Rating:** 6
**Confidence:** 3

**Summary:**

This paper explores how DPO-style algorithms can leverage rating gap information as an additional signal. The authors propose methods including RDPO and ML-RDPO, which achieve superior statistical rates compared to standard DPO when ratings are sufficiently accurate, while demonstrating robustness to rating noise. Comprehensive experiments across various LLMs and benchmarks show consistent performance gains over existing DPO-based approaches.

**Strengths:**

1. The proposed methods demonstrate consistent improvements across diverse models and benchmarks.
2. RDPO/ML-RDPO provably achieve faster convergence than DPO under the Bradley-Terry model while maintaining robustness to rating noise.

**Weaknesses:**

1. While the proposed approach shows promise, the technical novelty could be further clarified. The method appears to draw heavily from distill-DPO and DPO.
2. In the 'Experiments to assess robustness' section, how do other baseline methods perform in terms of robustness?
3. Figure 4 indicates that ML-RDPO still heavily relies on rating information. Furthermore, comparisons with other DPO variants are absent, limiting the assessment of the method's unique advantages.

**Questions:**

1. Why does ML-RDPO consistently underperform RDPO in most scenarios shown in Figure 2?

---

> ### Author Response · Authors · 2025-11-15
>
> Thanks a lot for your positive review and for your interesting questions. We hope that our answers might lead you to an even more positive perception of our submission.
>
> ***While the proposed approach shows promise, the technical novelty could be further clarified. The method appears to draw heavily from distill-DPO and DPO.***
>
> We would like to bring to the reviewer's attention that there are important technical novelties compared to the DPO and distilled DPO papers.
>
> Indeed, notice that Distilled DPO paper focuses on learning only from ratings and not from ratings and ranking. Moreover, while in the Distilled DPO paper, the use of the squared loss was not justified, in our case, the squared loss follows from a maximum likelihood formulation, which is novel to our work.
>
> Furthermore, neither DPO nor Distilled DPO comes with a detailed sample complexity analysis comparable to ours.
> More in detail, the derivation and theoretical analysis of our methods rely on the following new techniques.
>
> ***RDPO Innovation***
>
> (1) The RLHF formulation, which optimizes a linear combination of rating and ranking reward (equation 8), is novel.
>
> (2) For what concerns the analysis, the novelty lies in bounding the estimation error of the linear combination of rewards. This is difficult because of the potential dependence between the rating and the ranking reward. This requires a careful covering argument, which we detailed on page 23 in the proof of Lemma C.2.
>
> (3) The use of the class $\Pi'$ in Theorem 4.1 is novel to our work. Notice that $\Pi'$ depends on the reward $\hat{r}$ and its capacity can be tuned by the tuning of $\beta$ and $\beta_1$. This innovation also suggested incorporating penalties in the RDPO loss as specified in Appendix F.5.
>
> ***ML-RDPO Innovations***
>
> (1) The maximum likelihood perspective used to derive the method never appeared before. Notice that no similar analysis can be found in the DPO or Distilled DPO paper.
>
> (2) The analysis relies on a new technique that lower bounds the Hellinger distance between the joint rating and ranking distributions in terms of a convex combination of the Hellinger distances between probabilities between ratings only and rankings only, respectively (see lines 1354-1360 and lines 1514-1521 for the analogous derivations in the heterogeneous case). To the best of our knowledge, this proof technique based on the information processing inequality has not been used before in the theoretical analysis of alignment methods.
>
> **In the 'Experiments to assess robustness' section, how do other baseline methods perform in terms of robustness?**
>
> We do not think of the robustness experiments in Figure 3 as a comparison with other models but rather as a numerical verification of our theory, which suggests that more robustness to rating noise should be observed for higher values of $\beta_1$.
>
> We think that a large-scale investigation focusing fully on the robustness of alignment models against the noise in the rating and in the ranking information deserves independent future work.
>
> ***Figure 4 indicates that ML-RDPO still heavily relies on rating information. Furthermore, comparisons with other DPO variants are absent, limiting the assessment of the method's unique advantages.***
>
> We think that the fact that ML-RDPO's performance depends on the rating is to be expected, and it witnesses that ML-RDPO can achieve our goal of leveraging the rating information on top of the ranking one.
> In the plot, we do not have other methods because there are no known methods (beyond ML-RDPO) that can be applied in situations where the ratings are available only for some responses.  Figure 4 aims to highlight this unique advantage of ML-RDPO.
>
> We included only the DPO line because DPO is the limit of ML-RDPO when no rating is available. Interestingly, we notice that ML-RDPO can obtain advantages over DPO even if only half of the dataset is labelled with ratings. In conclusion, let us reiterate that RDPO, RPO, and MAPPO require the rating information for each preference pair; therefore, those methods can not be applied in the experimental setup of Figure 4.
>
> ***Why does ML-RDPO consistently underperform RDPO in most scenarios shown in Figure 2?***
>
> It is difficult to respond with certainty, but we would like to note that the derivation of RDPO does not require the Gaussian and conditionally independent rating assumption needed by ML-RDPO. This fact might make RDPO more robust to practical experiments where the above assumptions might be violated.
> On the positive side, ML-RDPO can be applied in cases with missing ratings, as explained in the answer above. On the contrary, RDPO requires ratings for each pair of responses to be implemented.
>
> In conclusion, thank you for your positive review and interesting comments. We are happy to discuss more in case you have any remaining concerns.

---

### Official Review · Reviewer_VeC4 · 2025-11-01

**Soundness:** 3
**Presentation:** 2
**Contribution:** 2
**Rating:** 4
**Confidence:** 5

**Summary:**

This paper studies the usage of rating-gap information (the scalar difference in ratings of two responses) along with pairwise ranking data for LLM alignment.

The authors derive two main algorithms: Rating DPO (RDPO), which incorporates the rating gap into DPO, and Maximum-Likelihood Rating DPO (ML-RDPO), derived from a joint likelihood perspective.

Theoretical analyses regarding acceleration and robustness to noisy ratings are given, which i quite liked. It adds a theoretical grounding to the work. Experiments are conducted on Zephyr, Mistral, and Llama-3 base models evaluated on AlpacaEval and ArenaHard.

**Strengths:**

1. Sound Theoretical backing to Algorithmic Claims

The paper is theoretically sound. Several DPO extensions add heuristic regularization terms. Instead, the authors derive ML-RDPO from a joint maximum likelihood perspective. The statistical bounds provided in Theorem 4.1 serve as a theoretical sanity check, confirming that including **accurate** rating information theoretically improves convergence rates compared to ranking-only methods under ideal conditions. (Note the noise in ratings typically sourced from reward models (Which are stochastic) is standard, hence how much this assumption of rating accuracy holds is questionable).

---

2. The idea of using ratings alongside rankings is certainly a good one.

The idea of using ratings alongside the rankings is good. But the idea is not new. It was first proposed in InfoNCA in neurips 2024 [1] and then extended into the DPO loss function [2,3]. These are references in this area that would be useful to include in their literature review for completeness.


---

### References

[1] Chen, H., et al. (2024). Noise contrastive alignment of language models with explicit rewards. Advances in Neural Information Processing Systems, 37, 117784–117812.

[2] Sun, S., et al. (2025). Reward-aware preference optimization: A unified mathematical framework for model alignment. arXiv preprint arXiv:2502.00203.

[3] Gupta, T., et al. (2025). Multi-Preference Optimization: Generalizing DPO via Set-Level Contrasts. arXiv preprint arXiv:2412.04628.

**Weaknesses:**

1. No improvement on Old Baselines:

The proposed method fails to improve on SIMPO for Llama, the strongest baseline that they tested on. The method only shows gains on older models (like Zephyr-7B-beta).

---

2. Theoretical Robustness to noise

Theorem 4.1 is contingent on **knowing the noise level** to set the hyperparameter $\beta_1$ correctly. In practice, this may lead to the brittleness observed in Appendix F.4, where $\beta_1$ must be tuned by orders of magnitude (e.g., 0.1 vs 0.005) across different models. The theoretical guarantee of robustness hence is contingent on finding an appropriate $\beta_1$ suitable to any new model or dataset, is it not?

---

3. Please consider recent baselines.

The paper reports ~29% Win Rates on Llama-3-8B, and shows equal performance with SIMPO [Note Simpo's own paper shows higher WR% and LC-WR % -- see Table 1 of the paper]. But please see the references below which exceed these numbers. For instance, RSPO [1] reports ~35% *Length-Controlled* win rates, while other recent multi-preference and reference-free approaches [2, 3] have reported win rates exceeding 50% on comparable benchmarks. Furthermore Chen et al.,2024 [4] provide a loss which is close to ML-RDPO. Incorporating these variants may better contextualize the method's true competitiveness.


---

### References

[1] Tang, X., et al. (2025). Game-Theoretic Regularized Self-Play Alignment of Large Language Models. arXiv preprint arXiv:2503.00030.

[2] Gupta, T., et al. (2025). REFA: Reference Free Alignment with Fine-Grained Length Control. COLM 2025.

[3] Gupta, T., et al. (2025). AMPO: Active Multi Preference Optimization for Self-play Preference Selection. ICML 2025.

[4] Chen, H., et al. (2024). Noise contrastive alignment of language models with explicit rewards. Advances in Neural Information Processing Systems, 37, 117784-117812.

[5] Wu, Y., et al. (2025). Self-play preference optimization for language model alignment., International Conference on Representation Learning (Vol. 2025, pp. 91558–91582).

**Questions:**

*   **Degradation on Llama-3.1:** Why do RDPO and ML-RDPO not improve upon the simpler SimPO baseline on Llama-3.1-8B even though there is access to more information. ideally, given the rating information as well as ranking leads to performance improvement. Is it because of lack of sufficient tuning of the hyperparameters

*   **Gaussian Assumption:** Theorem 4.2 and the derivation of ML-RDPO rely on the assumption that rating gaps are Gaussian distributed. Can you verify this assumption empirically on your datasets (e.g., UltraFeedback)? I'm wondering if the real-world rating distributions may have some heavy tail-ness that might violate this. For example, a histogram of gaps on UF would be a great contribution to the community using this training setup.

---
### Technical Question

- Simpo paper's own numbers are higher than those reported in your paper (as per my understanding in the same setting). 40%LC and 37% WR see Table 1 of their paper. Any reason for this discrepancy?


---

### Suggestion

*   **Baselines:** Please consider extending your work with more recent algorithmic works in this area to make it more empirically competitive.

---

> ### Author Response · Authors · 2025-11-15
>
> Thanks a lot for your review and for pointing out the references we missed.
> We extended our literature review in the extended version to include those citations in Appendix A. The new text is highlighted in blue.
>
> Thanks in particular for pointing us to the improved version of SIMPO, which does not use $\pi_{\mathrm{ref}}$ in the loss function. However, we would just like to clarify that our goal is not to improve over all the baseline free methods but rather to present a principled and practical way of using rating information in LLM alignment. We therefore designed our experimental evaluation with this intent.
>
> Please find below answers to your concerns. We hope that these will lead you to reconsider your assessment and to a potential higher rating.
>
> ***The proposed method fails to improve on SIMPO for Llama, the strongest baseline that they tested on. The method only shows gains on older models (like Zephyr-7B-beta).***
>
> Please notice that RDPO improves over SIMPO when Llama is used as base model, the win rate vs GPT4 attained by SimPO is 27.5% in Alpaca Eval vs the 29.2% of our RDPO. The difference is even larger, in Arena-Hard Simpo attains a 31.8% winrate vs the 35.4 % of RDPO.
>
> ***Theorem 4.1 is contingent on knowing the noise level to set the hyperparameter
>  correctly. In practice, this may lead to the brittleness observed in Appendix F.4, where
>  must be tuned by orders of magnitude (e.g., 0.1 vs 0.005) across different models. The theoretical guarantee of robustness hence is contingent on finding an appropriate
>  suitable to any new model or dataset, is it not?***
>
> We agree that $\beta_1$ needs to be set knowing the noise level. Nevertheless, we think that Theorem 4.1 offers guarantees of acceleration-robustness tradeoff for the first time. It is an interesting open question to derive an algorithm that adapts automatically to the noise level.
>
> In practice, RDPO comes only with a single hyperparameter to tune, i.e. $\beta_1$, which appears to be robust both **across models** and **noise** levels. Notice in contrast that SIMPO requires tuning of two hyperparameters, the margin $\gamma$ and the $\beta$, which should be taken larger than the value used in DPO according to the original SIMPO paper.
>
> **Robustness across models** As we state in Appendix F.4, we tuned $\beta_1$ only for Zephyr-7B as base model and Alpaca-Eval as evaluation benchmark. The chosen value $\beta_1 = 1/10$ is kept fixed in all experiments in Figure 2, leading to strong results also for Llama or Mistral as base model.
>
> **Robustness across noise** We notice in Figure 3 that the chosen value of $\beta_1$ performs similarly for all levels of noise in the ratings. It suffers only a very modest decline in performance as the noise increases. We think that this experiment rules out the suspicion that $\beta_1$ should be retuned when the rating noise ( and hence the dataset) changes.

---

> > ### Author Response · Authors · 2025-11-15
> >
> > ***The paper reports ~29% Win Rates on Llama-3-8B, and shows equal performance with SIMPO [Note Simpo's own paper shows higher WR% and LC-WR % -- see Table 1 of the paper]. But please see the references below which exceed these numbers. For instance, RSPO [1] reports ~35% Length-Controlled win rates, while other recent multi-preference and reference-free approaches [2, 3] have reported win rates exceeding 50% on comparable benchmarks. Furthermore Chen et al.,2024 [4] provide a loss which is close to ML-RDPO. Incorporating these variants may better contextualize the method's true competitiveness***
> >
> > Before providing our answer, let us mention that we compared with SIMPO because it seems to be a popular baseline used in the literature. However, our focus is not to improve over all reference-free methods. Our goal is rather to present a principled way to incorporate the rating information in alignment.
> >
> > We think that the higher performance reported in the SimPO paper is very likely explained by the fact that the base model is different. Notice, indeed, that Table 1 of the Simpo paper reports results which are not comparable to ours: we use Llama-3.1-8B as base model while they use Llama-3.1-8B-Instruct.
> > As it can be seen here https://huggingface.co/blog/llama31#whats-new-with-llama-31 Llama-3.1-8B-Instruct already undergoes a RLHF training phase. Our preference was to compare alignment algorithms from a model that had not been aligned in the previous stages of the training.
> >
> >
> > Similarly, the results in the RSPO paper are not comparable to ours (i) because it is an online method while ours is offline, and (ii) they use a Mistral-7B-Instruct base model while we use only an SFT version.
> >
> > Moreover, we think that it is important to highlight that the above 50% winrate reported in [2,3] is not comparable to our experiments because (i) Instruct models are used as a starting point and (ii) online interaction is needed, while our methods are completely offline. Notice, indeed, that we used the ultrafeedback dataset without generating new responses according to our base model, which is instead done in [3] ( see their first paragraph on their page 8).
> >
> > In our opinion, a potential fair comparison between our results and the ones in the references you mentioned could be between our results in Figure 2 and the ones presented in [2] ( Table 1).
> > These experiments are both offline and use the same base model. We include the results of the comparison in the following. When Llama is the base model, our approach is superior and vice versa for Mistral.
> > |     Llama-3.1-8B   | RDPO (Ours) | REFA ( Best from [2])
> > |:-------|:-----------:|:-----------:|
> > | Alpaca-Eval | **29.2** | 24.2  |
> > | Arena-Hard | **35.4**  | 26.5 |
> >
> > |     Mistral-7B   | RDPO (Ours) | REFA ( Best from [2])
> > |:-------|:-----------:|:-----------:|
> > | Alpaca-Eval | 22.3 | **23** |
> > | Arena-Hard | 14.1  | **16.6** |
> >
> >
> > Finally, we did not manage to catch the similarities between the algorithms proposed in [4] and ML-RDPO. We are happy to discuss any relation we might have missed if the reviewer can provide a more precise pointer.
> >
> > ***Degradation on Llama-3.1: Why do RDPO and ML-RDPO not improve upon the simpler SimPO baseline on Llama-3.1-8B even though there is access to more information. ideally, given the rating information as well as ranking leads to performance improvement. Is it because of lack of sufficient tuning of the hyperparameters***
> >
> > We would like to bring to the reviewer's attention that, as shown in our top row of Figure 2, RDPO does improve over SimPO when Llama-3.1 is used as base model.

---

> ### Author Response · Authors · 2025-11-15
>
> ***Gaussian Assumption: Theorem 4.2 and the derivation of ML-RDPO rely on the assumption that rating gaps are Gaussian distributed. Can you verify this assumption empirically on your datasets (e.g., UltraFeedback)? I'm wondering if the real-world rating distributions may have some heavy tail-ness that might violate this. For example, a histogram of gaps on UF would be a great contribution to the community using this training setup.***
>
> We can not think of a way to verify the Gaussian assumption for ultrafeedback because we do not have access to the model (the rater) that was used to generate the chosen and rejected scores in the ultrafeedback dataset. Notice that our Gaussian assumption is on the distribution of the ratings *conditioned on the prompt and responses pair*. Therefore, to verify it, we should query the rater multiple times and report the scores on a histogram. Notice that the histogram of gaps on UF would be an empirical estimation for the distribution of the ratings *without conditioning on the prompt and responses pair*.
>
> We would like to conclude by highlighting that the Gaussian assumption is often very natural in many contexts. Moreover, we highlight that our method, derived by building on this assumption, offers a convincing performance in practice. This fact witnesses that the Gaussian assumption is probably a good model for the real conditional distribution of the ratings. Therefore, we think that the Gaussian assumption is well justified.
>
> Similarly, notice that it is difficult to verify that the Bradley-Terry assumption holds. However, using the BT assumption is widely accepted because the RLHF and DPO-like methods derived leveraging it have a strong practical performance. Therefore, the BT assumption is a good approximation of the true unknown ranking distribution. We kindly invite the reviewer to think about the Gaussian assumption in an analogous manner.
>
>
>
> ***Simpo paper's own numbers are higher than those reported in your paper (as per my understanding in the same setting). 40%LC and 37% WR see Table 1 of their paper. Any reason for this discrepancy?***
>
> Yes, as previously mentioned, we think that the win rates difference is very likely explained by the fact that the choice of the base model is different between the SIMPO paper and ours.
>
> **Conclusion**
> To conclude, we will add citations to the works we missed, i.e.. However, we do not necessarily feel that an experimental comparison with W-MPO, REFA, and RSPO is needed as there are some important differences in the setting:
> (1) MPO and W-MPO use multiple winners and looser answers; when only one pair is use,d they essentially reduce to DPO, which we include as a baseline. Similarly, REFA boils down to SIMPO when only a pair of responses (one chosen and one rejected) is available.
> (2) RSPO requires online generation of the responses
>
> Finally, we think that it is very interesting to observe that if in the derivation of W-MPO, the definition of $\Delta W_{\mathrm{abs}}$ is replaced with $\Delta W(y) = S(y) - S_{\mathrm{mean}}(x)$, and only two responses are available, then one would obtain our Ratings DPO.
>
> Thank you again for the time spent reviewing our paper. We are happy to participate in the discussion in case you have any remaining concerns.

---

> ### Author Response · Authors · 2025-11-26
>
> Dear reviewer,
>
> as the end of the rebuttal is approaching we wanted to kindly ask if our responses clarifies your concerns.
>
> In a very short summary, we think that the experimental setting of other works that you pointed us too is not comparable to ours because the reference models are different and most other methods are not offline.
>
> Moreover, we wanted to politely bring to the reviewer attention that AMPO and MPO are for multi preference which is not our setting.
>
> We are happy to engage in further discussion and we are thankful for the many references that you included in your review and that we cited in our revision ( see our extended literature review in Appendix A)
>
> Best,
>
> Authors

---

### Meta-Review · Area_Chair_owAB · 2026-01-03

**Summary:**

This paper studies how Direct Preference Optimization (DPO)–style alignment methods can leverage additional rating-gap information beyond pairwise rankings. The authors propose RDPO, RIPO, and ML-RDPO, provide theoretical sample-complexity analyses suggesting faster statistical rates under accurate ratings, and present experiments on AlpacaEval and ArenaHard across several base LLMs.

Reviewers generally found the problem well motivated and appreciated the principled theoretical derivations, particularly the maximum-likelihood formulation and robustness analysis. However, the overall feedback remained mixed to negative. Key concerns included: (i) unclear or overstated practical impact, as improvements over strong recent baselines (e.g., SIMPO and newer preference-based methods) are limited or inconsistent; (ii) strong and difficult-to-verify theoretical assumptions (e.g., Gaussian rating gaps, unknown noise ratios) that weaken the practical relevance of the guarantees; and (iii) narrow experimental scope, relying primarily on UltraFeedback and relative-comparison benchmarks without reference-based evaluations.

While the rebuttal clarified several misunderstandings, added citations, and resolved some theoretical questions, it did not adequately address the above issues. The average score remains below 5, and the reviewer consensus does not support acceptance.

**Reviewer Concerns:**

Concerns reasonably addressed:

•	Misunderstandings of motivation:
The authors clarified that the paper does not aim to solve likelihood displacement or OOD generalization, which led one reviewer to revise their assessment upward.

•	Theoretical derivations:
Reviewers acknowledged that the derivations are principled and that the assumptions are standard within DPO-style theory, with one reviewer explicitly raising their score after clarification.

•	Baseline comparability:
The rebuttal explained differences in base models (e.g., SFT vs. Instruct) and online vs. offline settings, partially addressing discrepancies with reported numbers in prior work.

Outstanding concerns:

•	Practical tuning and robustness:
Despite clarifications, reviewers remain unconvinced that the key hyperparameter controlling trust in ratings can be set reliably in new datasets or settings.

•	Limited empirical scope:
Experiments are largely restricted to a single training dataset (UltraFeedback) and two relative benchmarks, with no evaluation on reference-based tasks (e.g., instruction-following with gold answers).

•	Competitiveness against recent methods:
The paper does not convincingly demonstrate advantages over newer alignment methods that achieve substantially higher win rates under comparable or stronger settings.

•	Incremental contribution:
Several reviewers still view the methods as incremental extensions of DPO and distill-DPO rather than a clear step change.

**Reviewer Scores:**

•	Reviewer VeC4: Increased from 4 → 5 or 6 after rebuttal.

•	Reviewer 2n9a: Remains at 6, but explicitly stated acceptance is not necessary.

•	Reviewer zdGC: Increased from 2 → 3 or 4 after clarifications.

---

### Decision · Program_Chairs · 2026-01-26

Reject